# MicroRNA-1253 Regulation of WASF2 (WAVE2) and its Relevance to Racial Health Disparities

**DOI:** 10.3390/genes11050572

**Published:** 2020-05-20

**Authors:** Mercy A. Arkorful, Nicole Noren Hooten, Yongqing Zhang, Amirah N. Hewitt, Lori Barrientos Sanchez, Michele K. Evans, Douglas F. Dluzen

**Affiliations:** 1Department of Biology, Morgan State University, Baltimore, MD 21251, USA; meark1@morgan.edu; 2Laboratory of Epidemiology and Population Science, National Institute on Aging, Baltimore, MD 21224, USA; norenhootenn@mail.nih.gov (N.N.H.); amirah.hewitt@gmail.com (A.N.H.); lori.barrientossanchez@gmail.com (L.B.S.); EvansM@grc.nia.nih.gov (M.K.E.); 3Laboratory of Genetics and Genomics, National Institute on Aging, Baltimore, MD 21224, USA; yongqing.zhang@nih.gov

**Keywords:** hypertension, differential gene expression, microRNA, actin cytoskeletal regulators, endothelial cell, health disparities, African American, women, race

## Abstract

The prevalence of hypertension among African Americans (AAs) in the US is among the highest of any demographic and affects over two-thirds of AA women. Previous data from our laboratory suggest substantial differential gene expression (DGE) of mRNAs and microRNAs (miRNAs) exists within peripheral blood mononuclear cells (PBMCs) isolated from AA and white women with or without hypertension. We hypothesized that DGE by race may contribute to racial differences in hypertension. In a reanalysis of our previous dataset, we found that the Wiskott–Aldrich syndrome protein Verprolin-homologous protein 2 (WASF2 (also known as WAVE2)) is differentially expressed in AA women with hypertension, along with several other members of the actin cytoskeleton signaling pathway that plays a role in cell shape and branching of actin filaments. We performed an in silico miRNA target prediction analysis that suggested miRNA miR-1253 regulates WASF2. Transfection of miR-1253 mimics into human umbilical vein endothelial cells (HUVECs) and human aortic endothelial cells (HAECs) significantly repressed WASF2 mRNA and protein levels (*p* < 0.05), and a luciferase reporter assay confirmed that miR-1253 regulates the *WASF2* 3′ UTR (*p* < 0.01). miR-1253 overexpression in HUVECs significantly increased HUVEC lamellipodia formation (*p* < 0.01), suggesting the miR-1253–WASF2 interaction may play a role in cell shape and actin cytoskeleton function. Together, we have identified novel roles for miR-1253 and WASF2 in a hypertension-related disparities context. This may ultimately lead to the discovery of additional actin-related genes which are important in the vascular-related complications of hypertension and influence the disproportionate susceptibility to hypertension among AAs in general and AA women in particular.

## 1. Introduction

Throughout the United States, systemic arterial hypertension and hypertension-related conditions, including coronary atherosclerotic heart disease and cerebrovascular disease, have disproportionate incidence, mortality, and morbidity among African Americans (AAs). AA women are at particular risk. Between 2013 and 2016, 66% of AA females over ≥20 years had hypertension, compared with 41.3% of non-Hispanic white women, 41% of Hispanic women, and 36% of Asian women [1]. Reducing or eliminating hypertension is predicted to reduce cardiovascular disease (CVD)-related mortality in women by almost 40% [1,2]. While 75% of AA women are aware of having hypertension, only 26% of AA women were able to control their high blood pressure [1]. A deeper understanding of the underlying biological mechanisms associated with hypertension may help reduce the burden of this condition. 

Differential gene expression (DGE) can be linked with ancestry and can influence how individuals respond to environmental stimuli and exposures [3,4] and their susceptibility to chronic diseases, including cancer [5] and peripheral arterial disease (PAD) [6]. Investigations have shown that DGE can predict outcomes to medical procedures including heart transplants [7]. DGE patterns are also linked with sex and gender. We previously reported that there is substantial differential mRNA and microRNA expression of hypertension-related genes and pathways in peripheral blood mononuclear cells (PBMCs) between AA and white women with hypertension [8,9]. We observed that genes in canonical pathways related to hypertension, such as the renin-angiotensin (RAS) pathway, are expressed in reciprocal directions and that this is associated with race [9]. A follow-up analysis of these results identified that poly-(ADP-ribose) polymerase 1 (PARP-1), a DNA damage sensor protein involved in DNA repair and other cellular processes, was upregulated in hypertensive AA women compared with white hypertensive women and contributed to cellular response to inflammation [8]. AA women with PAD also have elevated levels of endothelial oxidative stress and circulating inflammatory biomarkers compared with AA men with PAD [6], and these differences may influence disease outcomes in AA women.

Understanding not only the significance of DGE patterns in hypertension and CVDs, but also the underlying genetic mechanisms that regulate these patterns, will help further our understanding of the biological basis of these conditions. Expression of hypertension-related genes can be regulated by ancestral genomic polymorphisms and expression quantitative trait loci (eQTL) [10,11], but this does not account for all differences previously observed. This suggests that alternative mechanisms also contribute to gene expression differences in different individuals. A possible contribution to variations in gene expression levels may arise from regulation from microRNAs.

MicroRNAs (miRNAs) are short (20–22 nucleotide), single-stranded RNAs that post-transcriptionally regulate protein expression by binding with target mRNA 3′ untranslated regions (UTRs) and inhibiting translation, often by degrading the target mRNA [12]. miRNA regulation of protein expression is integral to the proper functioning and health of the endothelial tissues of the vasculature, underlying smooth muscle layers, and vascular response to changes in shear stress [13,14,15]. Disruption of miRNA regulation of hypertension-related genes can lead to endothelial dysfunction [14,16,17]. We previously reported that nine miRNAs exhibit disease- or race-specific differential expression and we have identified and validated novel hypertension-related targets for eight of these miRNAs [8,9]. 

Here, we have reanalyzed our microarray dataset to further our understanding of DGE in hypertensive women in hypertension-related pathways [9]. We identified significant DGE among genes within the actin cytoskeleton signaling pathway between hypertensive AA and white women and we have validated the previously identified and hypertension-related miR-1253 as a novel regulator of WASP family Verprolin-homologous protein 2 WASF2 (also known as WAVE2), an integral member of the actin cytoskeleton pathway. 

## 2. Materials and Methods

### 2.1. Study Participants

We reanalyzed our previous microarray dataset [9] of age-matched African American and white females who were either hypertensive (HT) or normotensive (NT) and were previously chosen from the Healthy Aging in Neighborhoods of Diversity across the Life Span (HANDLS) study of the National Institute on Aging Intramural Research Program (NIA IRP) of the National Institutes of Health (NIH) [18]. The demographics and clinical information for this re-examined cohort were previously described in extensive detail in [9]. The IRB of the National Institute of Environmental Health Studies, NIH, approved this study and all participants signed written informed consent. 

### 2.2. Microarray, Target Prediction, and Pathway Analysis

Gene expression levels within peripheral blood mononuclear cells (PBMCs) featured in this study were previously analyzed and quantified using the Illumina Beadchip HT-12 v4 (San Diego, CA, USA) as described in [9] and can be found in the Gene Expression Omnibus (GSE75672). In this current study, gene expression in HAECs was analyzed using the Illumina Beadchip HT-12 v4 and RNA was prepared and labeled according to the manufacturer’s protocol. Data were analyzed as previously performed [8] and outlying technical replicates were removed. Raw signals were analyzed by Z-score normalization [19] and individual genes with an average intensity > 0, false discovery rate < 0.3, *p*-value < 0.05, and fold change >|1.5| were considered significant and these HAEC microarray datasets can be found in the Gene Expression Omnibus and will include our miR-1253 datasets (GSE139286). Gene expression data, including Z-ratio and fold change, were imported into Ingenuity Pathway Analysis (IPA; Ingenuity Systems, Redwood City, CA, USA) and we used default and custom settings to perform pathway analyses of genes significantly affected by miR-1253 overexpression and compared with a scrambled negative control. DIANA-microT v5.0 with an applied filter of 0.5 [20] and TargetScan v7.2 [21] were used for miR-1253 target prediction. 

### 2.3. Cell Culture and Transfection

Primary human umbilical vein endothelial cells (HUVECs) were purchased and verified from Lonza and grown in EMB media supplemented with EGM- SingleQuot Kits (Lonza; Walkersville, MD, USA). Primary human aortic endothelial cells (HAECs) were purchased from Lonza and grown in EMB-2 media supplemented with EGM-2 SingleQuot Kits (Lonza). Cells were transfected with miR-1253 Pre-miR miRNA Precursor (Assay ID #PM13220) or scrambled Pre-miR miRNA Precursor negative control #1 (Catalog #AM17110) (ThermoFisher, Waltham, MA, USA). Mimics were transfected with Lipofectamine 2000 (ThermoFisher, Waltham, MA, USA).

### 2.4. The 3′ UTR Luciferase Reporter Assays

Two miTarget miRNA 3′ UTR plasmids were purchased from GenoCopeia (Rockville, MD) containing either the first (Catolog #Hmi088372a-MT06) or second halves (Catalog #Hmi088372b-MT06) of the *WASF2* 3′ UTR RNA sequencing. The miTarget plasmid vector (pEZX-MTO6) contains a luciferase reporter gene with attached 3′ UTRs of interest and downstream renilla luciferase for transfection efficiency controls. HUVECs were co-transfected with 50 ng of either *WASF2* 3′ UTR plasmid and with either 50 nM scrambled negative control or miR-1253 precursors using Lipofectamine 2000. Forty-eight hours later, luciferase and renilla activities were measured using the dual-luciferase reporter assay system (Promega) according to the manufacturer’s instructions. Renilla served as an internal transfection control and the ratio of luciferase/renilla was normalized to the scrambled control. All luciferase assays were measured using a Synergy HT Microplate Reader (BioTek, Winooski, VT) and performed in triplicate.

### 2.5. RNA Isolation and RT-qPCR

Total RNA was isolated from HAECs and HUVECs using TRIzol Reagents (ThermoFisher) with phenol/chloroform extraction according to the manufacturer’s protocol. RNA integrity was measured with a Nanodrop 2000 and cDNA was synthesized using random hexamers and Super Script II reverse transcriptase (Invitrogen, Carlsbad, CA). miRNA cDNA was synthesized using the QuantiMiR RT Kit and the provided universal reverse primer (Systems Biosciences, Mountain View, CA). All RT-qPCR reactions were performed with 2× SYBR green master mix (ThermoFisher) on either an Applied Biosystems model 7500 real-time PCR machine or a QuantStudio 6 Flex. miR-1253 levels were normalized to *U6* and *WASF2* levels were normalized to the average of *GAPDH* and *ACTB*. The following primers (forward and reverse) were used for each gene: miR-1253 forward 5′-AGAGAAGAAGATCAGCCTGCA-3′; U6 forward 5′-CGCAAGGATGACACGCAAATTC-3′; *WASF2* forward 5′-GCAGCATTGGCTGTGTTGAA-3′ and reverse 5′-GGTTGTCCACTGGGTAACTGA-3′; *ACTB* forward 5′-GGACTTCGAGCAAGAGATGG-3′ and reverse 5′-AGCACTGTGTTGGCGTACAG-3′; *GAPDH* forward 5′-GCTCCTCCTGTTCGACAGTCA-3′ and reverse 5′-ACCTTCCCCATGGTGTCTGA-3′. Gene expression levels were calculated using the 2-ΔΔCt methodology [22]. 

### 2.6. Western Blot Analysis

HAECs and HUVECs were washed 2x with cold PBS and then lysed in 2x Laemmli sample buffer on ice. Protein lysate was then loaded into a 10% polyacrylamide gel and separated. Protein levels were determined by anti-WASF2 (sc-373889; Santa Cruz Biotechnology, Dallas, TX, USA), anti-GAPDH (c-32233; Santa Cruz, CA, USA), and anti-ACTB (sc-1616; Santa Cruz, CA, USA) antibodies. Densitometry was performed using ImageJ software [23]. 

### 2.7. Immunofluorescence and Scoring of Cells with Lamellipodia and Filopodia

HUVECs were fixed in formaldehyde on glass slides and permeabilized in Triton-X. Cells were stained with rhodamine phalloidin (1:300) (Life Technologies), then with DAPI (1:10,000) and then mounted using ProLong (ThermoFisher Scientific). HUVECs were scored positive for the presence of lamellipodia if they displayed at least one actin-rich (phalloidin-positive) ruffled structure at the edge of the cell. Filopodia were scored positive if at least two actin-positive finger-like protrusions were observed emanating from the cell [24,25,26,27]. We used a Zeiss Observer D1 microscope with an AxioCam1Cc1 camera. Only cells that were either isolated or only attached to one other cell were counted. The number of positive cells is shown as a ratio to all DAPI-stained cells and cell area was measured using AxioVision Rel 4.7 software. This approach was modified from [24,25,26,27].

### 2.8. Statistical Analysis

The student’s *t*-test was used when comparing two groups unless otherwise indicated. A *p*-value of <0.05 was considered statistically significant and calculations were performed in Prism GraphPad v8.2.0, unless otherwise indicated.

## 3. Results

We sought to identify and validate novel hypertension-related targets for miR-1253, which was previously found to be significantly downregulated in PBMCs of hypertensive African American (AA) women [9], but had remained predominately unexplored in our prior analyses. Identifying hypertension-related targets for miR-1253 may increase our understanding of DGE in AA women with hypertension. In this current study, we used the DIANA-microT v5.0 [20] and TargetScan v7.2 [21] algorithms to update and expand upon our prior preliminary target analysis of miR-1253 [9] and validate potential miR-1253 mRNA targets. We used both algorithms to be as comprehensive in our target identification as possible, as both programs predict possible miRNA targets by emphasizing and weighing prediction criteria differently. DIANA-microT predicted 4972 mRNAs as potential targets and TargetScan identified 5345 mRNAs (Figure 1A, see Appendix A for complete list). There were 3088 unique mRNAs that overlapped between both prediction programs and we used this list moving forward with our in silico analysis. In order to pare down this gene list to identify novel miR-1253 targets related to hypertension, we compared the predicted 3088 miR-1253 target mRNAs with our previously curated list [9] of 1266 genes related to hypertension and inflammation. There were 200 predicted miR-1253 targets that overlapped with the hypertension-related gene set (Figure 1B; Appendix A).

Next, we compared the 200 predicted hypertension-related miR-1253 targets and compared this with our previously reported list [9] of 3554 mRNAs found to be significantly and differentially expressed in PBMCs in AA and white hypertensive women. This was done to further pare down this predicted target list and identify mRNAs previously known to have unexplored differential expression patterns in hypertensive women. We found that 117 of the predicted miR-1253 targets exhibited differential expression in PBMCs of hypertensive women (Figure 1C; Appendix A). We continued with this parsed list of the predicted miR-1253 targets for the remainder of our analysis as they were hypertension related and differentially expressed in AA or white women. To be as comprehensive as possible and capture all potential miR-1253 targets, particularly transcripts that are regulated by multiple mechanisms beyond miR-1253, we did not filter these 117 predicted targets by expression directionality in hypertensive women.

We next sought to further parse down this list of 117 mRNA targets and validate the role of miR-1253 in potentially regulating expression of some of these mRNAs. We overexpressed 50 nM of miR-1253 mimic in human aortic endothelial cells (HAECs) for 48 h and performed a discovery microarray to assess gene expression level changes. We used Ingenuity Pathway Analysis (IPA) [28] to identify the top diseases and disorders and molecular and cellular functions associated with miR-1253 overexpression. We observed that pathways related to cardiovascular disease, cellular growth and proliferation, and cellular assembly and organization were the most significantly affected in response to miR-1253 expression and within the top five of pathways in each category (Figure 1D). 

We next examined DGE in the actin cytoskeleton signaling pathway in our hypertension cohort by reanalyzing our previous microarray dataset GSE75672 from [9] (gene list in Appendix A). We chose this pathway for several reasons. First, given the role of actin cytoskeletal remodeling and signaling in hypertension and endothelial function [29,30,31]. Second, given the importance of this pathway in the cardiovascular diseases, cellular growth and proliferation, and cellular assembly and organization categories identified in our IPA analysis (Figure 1D, bold), in which 23 genes in the actin cytoskeleton pathway are found in the combined molecules list in IPA after removal of overlapping molecules (see Appendix A for full list of molecules and overlapping genes). Third, our prior analysis that RHOA, a member of the actin cytoskeleton pathway and associated with hypertension etiology in endothelial and vascular smooth muscle cells, was differentially expressed in PBMCs between AA and white hypertensive women. Fourth, the fact that this particular pathway in PBMCs is relatively unexplored with respect to hypertension. 

We used IPA to overlay our prior mRNA dataset of gene expression in 24 age-matched females who were either African American normotensive women (AANT), African American hypertensive women (AAHT), white normotensive women (WNT), or white hypertensive women (WHT; n = 6/group, as previously extensively described in [9]) to further examine DGE in the actin cytoskeleton signaling pathway. While only *PAK* was significantly higher in AANT compared with WNT in this pathway (Appendix A), we found that 27 genes of the 75 in the actin cytoskeleton signaling pathway are significantly different (*p* < 0.05 and |fold change| >1.5; Appendix A) when comparing AAHT with WHT (Figure 2). There are only three genes significantly different in this pathway between WHT and WNT in our cohort (*ARP2*, *ACTG1* (shown as F-actin in figure) and *SRC*; Appendix A) and *ARP2* and *ACTG1* are reciprocally expressed when comparing AAHT with AANT (Appendix A), suggesting that these genes exhibit DGE by race in hypertensive women. We also observed that there are more genes significantly different when comparing AAHT with AANT (Appendix A) than when comparing WHT with WNT, suggesting that the actin cytoskeleton signaling pathway is an overlooked gene pathway when examining health disparities in hypertension, particularly in AA women.

Microarray gene expression fold changes in PBMCs isolated from AANT, WNT, AAHT, and WHT were imported into Ingenuity Pathway Analysis (IPA) and overlaid onto the actin cytoskeleton pathway. In this figure, red indicates significantly upregulated expression and green indicates significant downregulation in AAHT compared with WHT. Grey indicates a non-significant difference and white indicates no data available. Nodes with green and red represent the differences for those nodes with two genes functionally interacting or genes with transcript variants. All fold changes and P-values are listed for each gene and each comparison in Appendix A and transcript variant information is available at GSE75672. AANT: African American normotensive women; AAHT: African American hypertensive women; WNT: white normotensive women; WHT: white hypertensive women.

In order to determine whether miR-1253 might play a role in the differential expression of genes within the actin cytoskeleton signaling pathway, we compared the list of mRNAs significantly downregulated in HAECs via overexpression of miR-1253 mimic against the list of 117 predicted miR-1253 targets which were differentially expressed in hypertensive women (Figure 1C). There were 747 mRNAs significantly repressed > 1.5-fold compared with the scrambled negative control (*p* < 0.05; FDR < 0.20; *n* = 5; Appendix A). Of these 747, 23 mRNAs overlapped with our list of 117 predicted targets and which were differentially expressed in hypertensive women (Table 1). When the list of 23 overlapping mRNAs was compared with the 75 genes in the actin cytoskeleton pathway, one of these genes, WASP family Verprolin-homologous protein 2 WASF2 (also known as WAVE2), was found to be differentially expressed between AAHT and WHT women (Figure 2, circled in purple). WASF2 is known to regulate actin cytoskeleton branching [27,32]. miR-1253 was also predicted to target two other genes in the actin cytoskeleton pathway, Filamin A, α (FLNA) and Ras Homolog A (RHOA). However, neither of these two mRNAs were significantly downregulated by miR-1253 in our screen. Therefore, we focused on WASF2 as a potential target of miR-1253. 

We performed a luciferase reporter assay using miTarget reporter vectors to confirm that miR-1253 can regulate the 3′ untranslated region (UTR) of *WASF2*. The 3′ UTR of *WASF2* is 3959 nucleotides in length and was split between two miRTarget plasmids. These heterologous reporter plasmids contain luciferase with a downstream renilla luciferase (RL) transfection control. The miRTarget *WASF2* 3′ UTR-1 plasmid contains the first 2010 nucleotides of the *WASF2* 3′ UTR, including the last 21 nucleotides of its coding region. The miRTarget *WASF2* 3′ UTR-2 plasmid contains nucleotides 1888 to 3959 of the *WASF2* 3′ UTR and there is a common overlap of 122 nucleotides of the 3′ UTR between plasmid 1 and 2 (Figure 3A). DIANA-microT predicted that miR-1253 could bind with the *WASF2* 3′ UTR at two positions, with seed sequences beginning at nucleotides 1617 (referred as binding site #1) and 1775 (binding site #2) (Figure 3A,B). Both DIANA-microT and TargetScan predicted that miR-1253 binds to the *WASF2* 3′ UTR at nucleotides 3734 to 3756 in the second half of the *WASF2* 3′ UTR, which is referred to as the WASF2 3′ UTR binding site #3 (Figure 3A,B). Human umbilical vein endothelial cells (HUVECs) were co-transfected with 50 nM miR-1253 or scrambled control mimics and either miRTarget *WASF2* 3′ UTR-1 or 3′ UTR-2. We observed significant repression of luciferase activity for miRTarget 3′ UTR-1 (*p* < 0.01, *n* = 3) and miRTarget 3′ UTR-2 (*p* < 0.001, *n* = 3) in the presence of miR-1253 and compared to a scrambled control (Figure 3C). These data indicated that miR-1253 can regulate the *WASF2* 3′ UTR and reduce protein expression.

We next validated whether miR-1253 can regulate WASF2 expression *in vitro*. We overexpressed the 50 nM miR-1253 mimic for 48 h in human aortic endothelial cells (HAECs). In the presence of miR-1253, *WASF2* mRNA levels were significantly repressed nearly 50% (*p* < 0.05; *n* = 3) and the corresponding WASF2 protein levels were significantly downregulated by nearly 60% (*p* < 0.01; *n* = 3) compared with a scrambled control mimic (Figure 4A). In order to verify that this is not a cell line-specific effect, we also performed the same experiments in HUVEC cells. miR-1253 mimics significantly repressed *WASF2* mRNA 55% (*p* < 0.01; *n* = 5) and WASF2 protein 38% (*p* < 0.001; *n* = 5) (Figure 4B). Together, these results confirm our in silico prediction that miR-1253 can regulate the expression of WASF2 protein in endothelial cells.

Given that WASF2 is a key regulator of actin cytoskeleton dynamics, we assessed whether this regulatory network may affect the actin cytoskeleton. We transfected 50 nM scrambled control or miR-1253 mimics into HUVECs for 48 h and stained with rhodamine phalloidin to visualize actin cytoskeletal structures. We observed morphological changes in cells transfected with miR-1253 mimic compared to scrambled control mimics (Figure 5). Protrusive actin-containing structures such as lamellipodia or filopodia are formed at the leading edge of cells. Lamellipodia form larger actin-containing ruffles, while filopodia are characterized by actin-containing finger-like extensions from the cell [33,34]. These structures can be identified by immunofluorescence staining of cells for actin. Cells with transfected miR-1253 had increased lamellipodia formation, as shown by concentrated actin-rich membrane ruffling at the edges of cells. Therefore, we scored these cells by the presence of either lamellipodia or filopodia. We observed that there was a significant increase in lamellipodia in HUVECs transfected with miR-1253, indicating an increase in actin-rich membrane ruffling at the edges of the cells (*p* < 0.001; *n* = 3 independent experiments) (Figure 5A,B). miR-1253 did not affect the formation of actin-rich filopodia projections. We did observe an increase in cell surface area of approximately 60%. However, this was not statistically significant (*p* = 0.09; *n* = 3 independent experiments) (Figure 5C). Together, miR-1253 regulates WASF2 in endothelial cells, leading to changes in endothelial cell lamellipodia formation. 

## 4. Discussion

Together, reanalysis of unexplored pathways within our prior data examining DGE patterns in AA and white women with hypertension indicates that a large number of genes within the actin cytoskeleton signaling pathway are differentially expressed between AA and white hypertensive women. Importantly, nearly all of these genes exhibit similar expression levels between normotensive AA and white women (Figure 2, Appendix A). This suggests that the DGE patterns associated with hypertension occur around the time the disease process begins or during and after sustained exposure to elevated systemic blood pressure levels. Previously, we found similar patterns in additional pathways related to hypertension [8,9] and this study provides further evidence that DGE is associated with individuals with high blood pressure. Here, we found that miR-1253, identified in our previous analysis [9] but without a functionally validated role in hypertension, was predicted to target *WASF2* in the actin cytoskeleton pathway. Our gene expression analysis in PBMCs led us to identify that miR-1253 can regulate the *WASF2* 3′ UTR, repress WASF2 protein expression in endothelial cells, and influence actin cytoskeletal dynamics with respect to increased lamellipodia formation in endothelial cells (Figure 4 and Figure 5). 

DGE within the actin cytoskeleton signaling pathway in hypertension has previously remained relatively unexplored, particularly in the context of AA women with hypertension. Most studies have examined the role of this pathway in downstream conditions—of which, hypertension is a major risk factor. Pathway analysis of gene expression in coronary artery atherosclerosis plaques identified that focal adhesion and actin cytoskeleton pathways as some of the most differentially expressed between early and late-stage plaques [35]. In human macrophages, FLNA expression is higher in advanced atherosclerotic plaques compared with intermediary plaques and inhibition of FLNA expression in mice reduced plaque development, suggesting a role for this gene and the actin cytoskeleton in hypertension-related CVDs [36]. FLNA is expressed in human and mouse endothelial cells after myocardial infarction. When its expression is inhibited, the endothelial response to cardiac repair, migration, and VEGF-A secretion was reduced, and this promoted left ventricular dysfunction and heart failure [37]. In our analysis, we found that AA women have lower levels of *FLNA* compared with white women, which may suggest that FLNA and its expression in specific contexts is relevant in hypertension etiology. 

Altered levels of other members of the actin cytoskeleton signaling pathway have been observed but not in the context of gender or race. Bradykinin receptors 1 and 2, which act as upstream regulators of vessel wall remodeling, are significantly upregulated in peripheral monocytes of essential hypertensives and hypertension treatment reduces their expression [38]. The Rho/ROCK signaling cascade regulates organization of the actin cytoskeleton and cell morphology, including adhesion of cells along the endothelium of the vasculature [39,40,41]. Members of the RhoA family have been extensively examined as targets for hypertension therapy [42], and given its upregulation in AA women with hypertension (shown here and in [9]), the targeting of elevated RHOA expression and the downstream impact on cytoskeleton function may be a novel area for intervention in AA women. Follow-up studies are warranted to investigate this. 

We identified differential expression of *WASF2* between AA and white women with hypertension. WASF2 is an actin nucleation-promoting factor and binds with the actin-related protein (Arp) 2/3 complex to promote actin filament nucleation and branching [43,44]. Variation in WASF2 expression modulates actin branching and influences the formation of cellular filopodia and lamellipodia [27,32,44,45,46]. We observed that repression of WASF2 levels due to miR-1253 overexpression increased the formation of lamellipodia and membrane ruffling, consistent with lamellipodia formation and actin elongation dynamics related to WASF2 expression modulation [32]. It is possible that miR-1253 regulation of WASF2 in hypertensives may influence endothelial integrity and lead to downstream complications, and additional studies are warranted to investigate this. 

Endothelial response to increased shear stress and laminar flow has been found to be associated with race. HUVECs isolated from AAs are more responsive to laminar shear stress compared with HUVECs from whites, including in pathways related to nitric oxide synthase and oxidative stress response. Importantly, in both cases, exercise was able to improve upon those changes [47,48]. A recent meta-analysis of studies comparing arterial stiffness between AAs and whites identified significant differences in AAs in aortic femoral pulse wave velocity and carotid-femoral pulse wave velocity [49] and build off of previous analysis that AAs can have impaired microvascular dilatory response [50]. Our findings here indicate that the actin cytoskeleton could influence or associate with these clinical observations and further consideration of the involvement of WASF2, miR-1253, and related pathway genes will be important to identify any direct roles. 

Modulation of WASF2 expression by miR-1253 in circulating PBMCs may contribute towards hypertension-related changes in membrane physiology and morphology and downstream complications, such as atherosclerosis. Follow-up studies will be necessary to ascertain this. Recent findings suggest that B cells [51] and several T cell subtypes can influence angiotensin II (AngII)-signaling and downstream cytokine release and inflammatory response in hypertensives [51,52,53,54]. M-positive monocytes mediate AngII-induced hypertension and promote downstream vascular dysfunction in response to elevated blood pressure [55]. While there have been few studies directly investigating the role of the actin cytoskeleton structure in these cells as a causative mechanism of hypertension, several members of this pathway, including RHOA and ROCK, have identified roles in T and other immune cell cytoskeletal structure [56,57]. This includes regulatory roles in lamellipodial function required for transendothelial migration during inflammatory response [56,58]. Our findings indicate that WASF2 may also be a player in immune cell cytoskeletal dynamics and future studies will help elucidate this contribution to any additional hypertension-related mechanisms. 

Previously, miR-1253 expression levels in plasma have been linked with glaucoma [59] and it is epigenetically silenced via hypermethylation in medulloblastoma tissues. [60]. miR-1253 regulates the expression of the long, non-coding RNA *FOXC2-AS1* in prostate cancer cells [61] and *WNT5A* in lung carcinoma [62]. Oncogenic circular RNA *circ_0030235* can act as a sponge of endogenous miR-1253 in pancreatic cancer cells and promote cellular growth and migration [63]. miR-1253 can also regulate *FOXC2-AS1* in vascular smooth muscle cells (VSMCs) to decrease cellular proliferation and miR-1253 plasma levels were significantly reduced in patients with atherosclerosis [64]. This is the only study currently linking miR-1253 with VSMCs, atherosclerosis, or any other cardiovascular disease. 

It is interesting to speculate how AA women with hypertension may have lost expression of miR-1253 in PBMCs. Considering that the miR-1253 loci is sensitive to hypermethylation in cancer [60], this could be a potential mechanism repressing miR-1253 expression levels in hypertensives. As well, ancestral polymorphisms can function as expression quantitative trait loci (eQTL) in macrophages and immune cells to influence differential gene expression and immune and metabolic function [3,65]. It is possible that miR-1253 expression is regulated by eQTLs in AA hypertensives and this will need to be explored further, as this would also provide a more thorough understanding if miR-1253 expression changes before, during, or after the development of essential hypertension. 

Many miRNAs play an important role in the normal and disease physiology of the vasculature. For example, miR-155 regulates endothelial eNOS and downstream vasodilation in human mammary arteries [66] and its expression is inversely correlated with target AGTR1 expression in untreated hypertensives [67]. Several miRNAs, including miR-143 and miR-145, regulate vascular smooth muscle cell function and have been found to be differentially expressed in PBMCs and correlated with 24 hr diastolic blood pressure and pulse pressure in individuals with hypertension [68]. It is unknown whether these miRNAs are correlated with disparities in hypertension, particularly in AA women, or involved in similar pathways as miR-1253. 

Our approach was useful in being able to examine DGE patterns in understudied populations and identify novel regulators of some of those genes. However, our study is limited because it is not known whether differential expression of miR-1253 or *WASF2* in AA women with hypertension is a contributing cause or a consequence of elevated high blood pressure. It is likely not a major cause of hypertension development, but given the pathways these genes are involved with, they should be explored further when it comes to understanding how to better manage or treat high blood pressure. We were also not able to ascertain whether miR-1253 is the major regulator of *WASF2* transcript levels in PBMCs or acting as a contributing role player. As the connections between circulating cells and the development and pathology of hypertension become clearer, these genes should be considered. The miR-1253–WASF2 interaction may also be relevant, or perhaps even more relevant, in tissues already known to be more directly related to the constriction and dilation of the vasculature, including the endothelial and smooth muscle layers. Our finding highlights the need to validate whether miR-1253 regulates WASF2 expression in primary endothelial or VSMCs, particularly in projects focused on examining mechanisms of hypertension-related disparity and in biospecimens from these populations. Given that there are no data in the literature examining whether changes in miR-1253 influence endothelial dysfunction in response to increased blood pressure, our study lays groundwork for these intriguing projects.

Together, we have identified the actin cytoskeleton as a possible avenue to explore to further our understanding of how hypertension may develop or present in different populations. Future studies will need to examine the miR-1253–WASF2 relationship in order to further elucidate their role in the development or pathology of hypertension.

## Figures and Tables

**Figure 1 genes-11-00572-f001:**
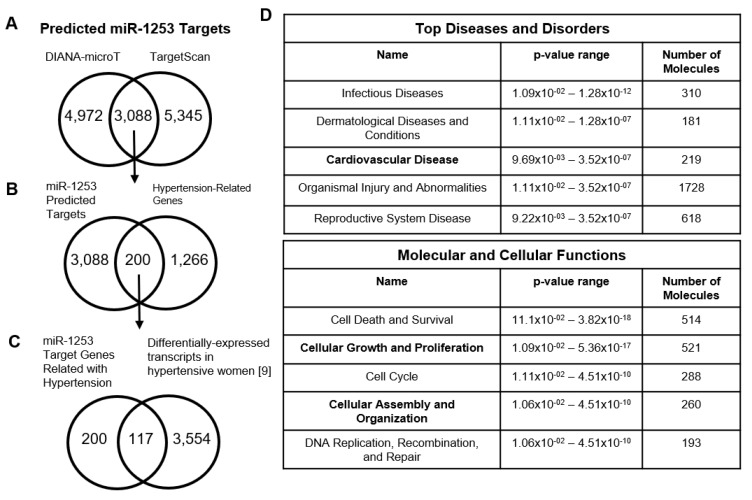
Target prediction analysis for miR-1253. (**A**) Venn diagram of the predicted miR-1253 targets overlapping between the DIANA-microT and TargetScan algorithms. (**B**) Venn diagram of overlapping, predicted miR-1253 targets that are within a previously curated [9] list of hypertension-related genes. (**C**) Venn diagram of the predicted hypertension-related miR-1253 targets that are significantly, differentially expressed in peripheral blood mononuclear cells (PBMCs) in hypertensive women. (**D**) List of significant top diseases and disorders (top) and molecular and cellular functions (bottom) in human aortic endothelial cells (HAECs) transfected with 50 nM miR-1253 mimic. These pathways were identified by Ingenuity Pathway Analysis.

**Figure 2 genes-11-00572-f002:**
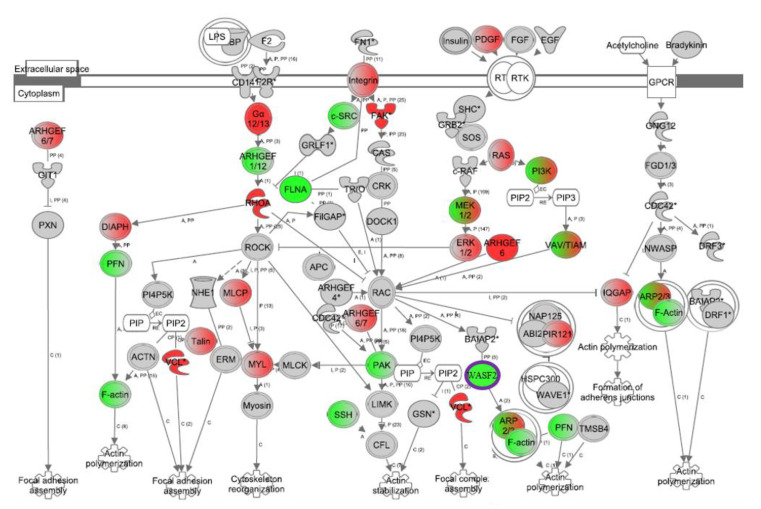
Gene expression analysis of the actin cytoskeleton in hypertensive women.

**Figure 3 genes-11-00572-f003:**
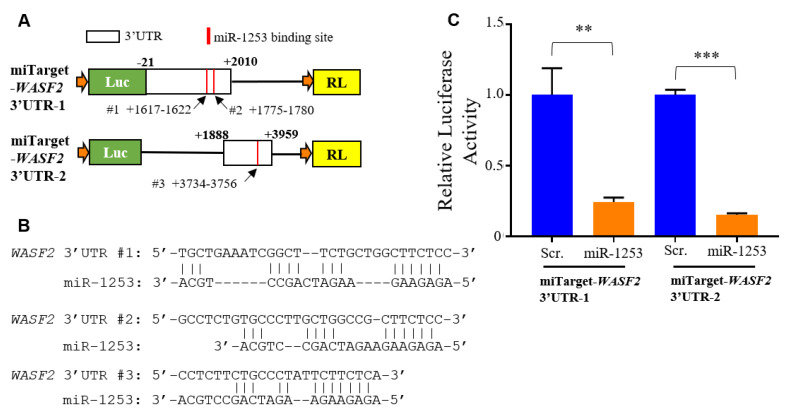
miR-1253 targeting of the *WASF2* 3′ UTR. (**A**) Schematic of the miRTarget *WASF2* 3′ UTR vectors (plasmid 1 and 2). The predicted binding sites of miR-1253 to the *WASF2* 3′ UTR are indicated in red with designated base pair positions. (**B**) Base pair schematic of binding site #3 of miR-1253 to the 3′ UTR region of *WASF2*, as predicted by TargetScan. (**C**) The relative expression of luciferase (Luc) reporter in the presence of 50 nM miR-1253 for 48 h and compared with scrambled control. Data were normalized to an internal renilla control and normalized to 1.0. ** *p* < 0.01; *** *p* < 0.001, by two-tailed student’s T-test.

**Figure 4 genes-11-00572-f004:**
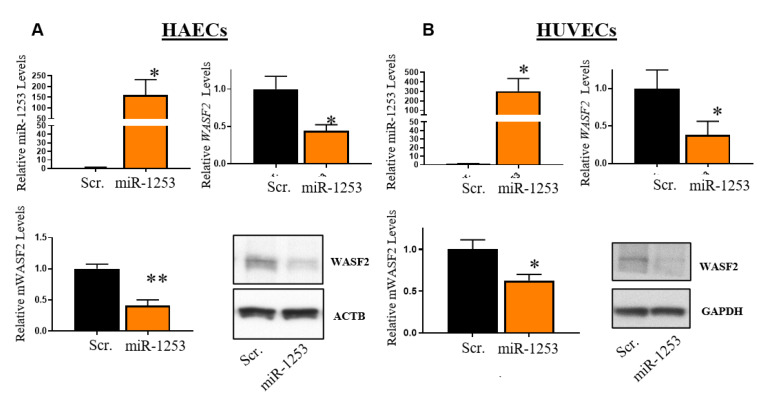
Overexpression of miR-1253 in HAECs and HUVECs reduces expression of WASF2. The 50 nM miR-1253 was transfected into HAECs (*n* = 3) (**A**) and HUVECs (*n* = 5) (**B**) for 48 h, and overexpressed in each cell line compared with a scrambled negative control mimic (scr.; top left). WASF2 mRNA expression was normalized to GAPDH in each cell line and shown relative to a scrambled control (scr.; top right). WASF2 proteins levels were normalized to β actin (HAECs) or GAPDH (HUVECs) and shown relative to a scrambled control (scr.; bottom left). Representative immunoblots are shown for WASF2 and loading controls in each cell line (bottom right); * *p* < 0.05, ** *p* < 0.01, by one-tailed T-test (for confirmation of miR-1253 and mRNA expression levels in each cell line) or two-tailed student’s T-test for all protein levels. Columns represent the mean ± S.D.

**Figure 5 genes-11-00572-f005:**
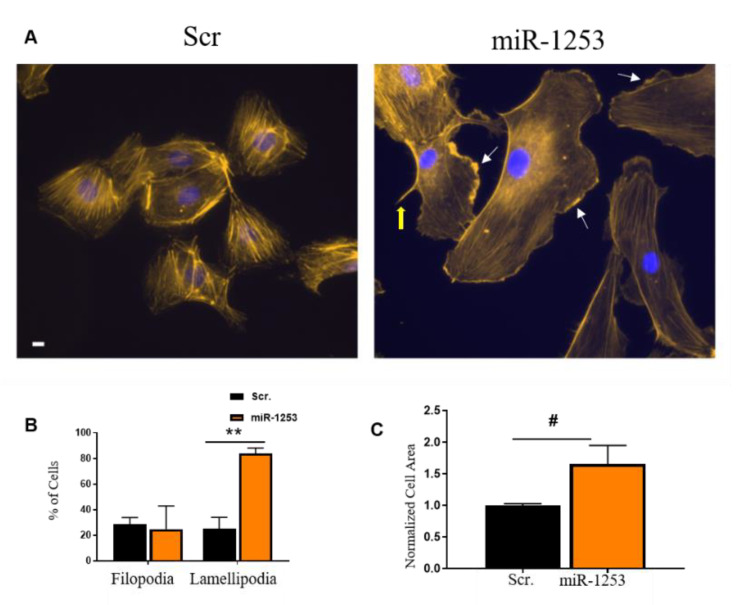
miR-1253 increased lamellipodia in HUVECs. (**A**) Representative pictures of HUVECs transfected with either scrambled control mimics (scr., left panel) or miR-1253 (right panel) and stained with rhodamine phalloidin for actin filaments and DAPI for nuclei visualization. White arrows indicate examples of scored lamellipodia and the thick yellow arrow indicates an example of scored filopodia. (**B**) Percent of cells visualized and counted for filopodia or lamellipodia in cells transfected with the scrambled or miR-1253 mimic versus total number of DAPI-stained cells (*n* = 3). (**C**) Quantitation of cell surface area of HUVECs transfected with the scr. control or miR-1253 mimic. ** *p* < 0.01, # *p* = 0.09; two-tailed student’s T-test. Scale bar = 10 µm. Columns represent the mean ± S.D. of three independent experiments and between a total of 28–89 (among scrambled conditions) and 25–46 cells counted (among mimic conditions) per experiment.

**Table 1 genes-11-00572-t001:** Summary of the predicted miR-1253 targets repressed in HAECs *.

Predicted miR-1253 Targets (Compared w/Scrambled Control)	FDR	Fold Change	*p*-Value	Z-Ratio
ABCB10	0.0172	−1.53	0.0032	−2.83
ACO1	0	−7.58	0	−11.94
ACSL1	0	−1.69	0	−3.32
DCUN1D5	0.0116	−1.87	0.002	−3.8
DPYSL2	0	−1.61	0	−2.47
DUSP14	0	−2.13	0	−4.46
MSN	0	−1.83	0	−3.15
PARP1	0	−4.4	0	−8.57
PDE12	0	−2	0	−4.36
POLA1	0	−1.53	0	−2.82
PTGER4	0	−1.72	0	−3.05
RAB27A	0	−2.06	0	−4.5
RSU1	0	−2.16	0	−4.78
RXRA	0	−1.54	0	−2.61
SEC62	0	−1.55	0	−2.96
SERINC3	0	−2.51	0	−5.53
SPARC	0	−1.91	0	−3.55
TFRC	0	−1.71	0	−3.1
TMEM127	0	−1.89	0	−4.07
TNS3	0	−1.83	0	−3.68
TOPBP1	0	−2.09	0	−4.35
UBE2N	0.0005	−1.56	0.0001	−2.67
WASF2	0	−1.64	0	−3.09

* mRNAs also found to be significantly and differentially expressed in PMBCs in hypertensive AA and white women and with hypertension-related pathways, outlined in [9].

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
