# Peer review of "MicroRNA-1253 Regulation of WASF2 (WAVE2) and its Relevance to Racial Health Disparities"

_genes, 2020, doi:10.3390/genes11050572_

Round 1

Reviewer 1 Report

The manuscript deals with racial differences regarding hypertension and, for this, it shows different lists of genes to conclude with the study of miR-1253 and the demonstration of WAVE2 as one of its targets. It is an interesting topic that needs further investigation but the present manuscript has some important problems.

  1. The title is not accurate because the molecular experiments presented here are not demonstrating the relevance of miR-1253 regulation of WAVE2 to hypertension. The experiments are performed on endothelial cells but there is no any hypertension model. In addition, maybe authors mean “racial disparities” instead of “health disparities”.

  1. The central point of the manuscript is the overexpression of miR-1253 to downregulate WAVE2 and see the effect on endothelial cells. But the basis of these experiments is in the findings of reference 9, by the same authors, in which a decrease of WAVE2 in the AAHT PBMCs is described, but also a decrease in miR-1253 in the same individuals (Figure 3 in reference 9). This would go against the proposition in this manuscript that the regulation of WAVE2 is by miR-1253, because miRNAs regulate genes in a negative way. It also contradicts that, in the present manuscript, miR-1253 is being overexpressed when what one would have to do to reproduce what happens in AAHT PBMCs is to inhibit it by antagomirs, for example.

  1. In Material and Methods section (lanes 93 to 97) it is not clear if Illumina Beadchips were used again to measure gene expression levels in PMBCs from the participants. It is more plausible that data obtained from reference 9 was considered. Indeed, the first paragraph of Results (from lane 159) does not contribute anything different from the reference 9. The 112 (111) genes are already in reference 9 predicted target genes for miR-1253 within the IPA gene sets and differentially-expressed in PBMCs. Even WAVE2 appeared in that reference with the name WASF2, the correct name, by the way. The new contributions of this manuscript begin in lane 183, with the in vitro experiments.

  1. In Material and Methods also, transfection experiments are not clear. Are miR-1253 pre-miRNA (precursors) or mimics (mature) used? Why 50 nM is used? Did you try different concentrations? Are transfections for luciferase assays performed also with Lipofectamine 2000?

  1. Supplementary File 1 is very confusing. Not all the lists of genes are in alphabetical order. Please, homogenize. Also, please review gene numbers in sets because, for example, the number of predicted targets that are also found in the list of 1,266 genes related to hypertension and inflammation is 111, not 112 (lane 167). In addition, these 1,266 genes are not only those related to hypertension and inflammation but those included in all IPA sets considered.

  1. In lane 201 it is stated that 75 genes are in the Actin Cytoskeleton signaling pathway but only 54 genes are shown in Supp File 1. Even, neither RHOA nor WAVE2 are in this list of genes. Maybe the correct list is the one in reference 9, with 84 genes, which includes WAVE2? In this column, what are the genes in 842-862 rows? It is very confusing.

  1. In lane 226, authors compare the 747 repressed mRNAs with the hypertension gene list. Why do they not with the Actin Cytoskeleton list? Considering that WAVE2 is in this last list.

  1. In lane 335, “AA women have higher levels of FLNA compared with white women” is true just in case of hypertensive women. In any case, FLNA has the same sign (negative) than WAVE2, for example, and WAVE2 is described here to have lower levels in AAHT. It is really confusing.

  1. The first paragraph of the discussion (from lane 312) includes basically results from the previous work of the authors but it is presented as if it had been obtained in the present manuscript. In addition, I cannot find sense to this sentence in lane 318: “further evidence that DGE is associated with individual gene expression levels in individuals with high blood pressure”.

  1. The second part of the statement (in lane 322) “… validate that miR-1253 can bind and regulate WAVE2 expression in endothelial cells and influence actin cytoskeletal dynamic” cannot be deduced from the results presented here.

  1. In lane 391, the use of bioinformatic analysis to identify and validate novel miRNA regulators for members of that pathway is not identified in the present manuscript and, the fact that WAVE2 could be validated for the authors as a miR-1253 target is not guarantee to validate other future bioinformatic findings.

Minor comments

  1. Gene names are mixed with protein names. The attempt to put gene names in italics is not useful if the official gene name is not used. WAVE2 should be WASF2.

  1. It is impossible to see Figure 2. Please, enhance it. But also labels in Figure 4 and Figure 5B and C.

  1. What does it mean “F-actin” in the list of genes? To my knowledge, F-actin is a polymer of actin, not a unique protein encoded by a gene. ACTG1 codes for gamma-actin.

  1. In lane 327, “plaques” instead of “plagues”.

  1. In lane 387, “It is unknown…”.

Author Response

Response to Review 1:

Dear Reviewer,

The authors thank you for providing a thorough and critical review our manuscript. We appreciate the opportunity to improve our work. Below, we have provided our responses and detailed the changes we have made to the manuscript to address these concerns. We thank the reviewer again for their time and hope this provides clarification.

Regards,

Doug Dluzen

  1. The title is not accurate because the molecular experiments presented here are not demonstrating the relevance of miR-1253 regulation of WAVE2 to hypertension. The experiments are performed on endothelial cells but there is no any hypertension model. In addition, maybe authors mean “racial disparities” instead of “health disparities”.

We thank the reviewer and we have updated our title to reflect this to “MicroRNA 1253 regulation of WAVE2 and its relevance to racial health disparities”.

  1. The central point of the manuscript is the overexpression of miR-1253 to downregulate WAVE2 and see the effect on endothelial cells. But the basis of these experiments is in the findings of reference 9, by the same authors, in which a decrease of WAVE2 in the AAHT PBMCs is described, but also a decrease in miR-1253 in the same individuals (Figure 3 in reference 9). This would go against the proposition in this manuscript that the regulation of WAVE2 is by miR-1253, because miRNAs regulate genes in a negative way. It also contradicts that, in the present manuscript, miR-1253 is being overexpressed when what one would have to do to reproduce what happens in AAHT PBMCs is to inhibit it by antagomirs, for example.

In this study, we primarily used our re-analysis of gene expression in hypertensive women as an avenue to explore differential gene expression in the actin cytoskeleton pathway and identify possible regulators of those genes. Indeed, our findings of miR-1253 repression in AA women with hypertension initiated this pursuit. We agree with the reviewer that it is unlikely miR-1253 is the primary expression regulator of WASF2 expression in PBMCs, but our experiment provides evidence that it contributes to a degree. microRNA expression of target mRNA transcripts is very dynamic and is very contingent upon the expression levels of the transcript, the microRNA, and the interplay of other regulatory factors related to the regulation of that specific transcript, often other microRNAs [1-3]. Because WASF2 expression is repressed in AA women with hypertensive, even the noted decrease in miR-1253 could still be an important factor. But we acknowledge that this is not investigated in our manuscript. Ideally, it would have been stronger data to see a reciprocal effect.

We have edited the discussion to emphasize that miR-1253 is a contributing regulator of WASF2 expression in PBMCs, and that it’s unknown if it’s the only. We added in the following sentences at lines 443-434:

“Our findings indicate that WASF2 may also be a player in immune cell cytoskeletal dynamics and future studies will help elucidate this contribution to any additional hypertension-related mechanisms.”

And lines 461-463:

“We were also not able to ascertain if miR-1253 is the major regulator of WASF2 transcript levels in PBMCs or acting as a contributing role player.”

We also agree that a study with antagomirs may be helpful in determining how the endogenous miR-1253 regulates WASF2 levels in cells, as this would likely be more representative of what is potentially occurring in PBMCs. However, at this moment, our laboratory is currently shutdown from the pandemic, as is those of our collaborators. This is an avenue we would like to pursue in the future, when we can, as well as in immune cell lines to be even more representative of gene expression in PBMCs. The use of miRNA over-expression in cell models is widely used as a means to validate the regulatory capacity of a microRNA and its predicted target. In this case, our over-expression studies were used to confirm that WASF2 mRNA and protein levels are repressed in the presence of miR-1253.

  1. In Material and Methods section (lanes 93 to 97) it is not clear if Illumina Beadchips were used again to measure gene expression levels in PMBCs from the participants. It is more plausible that data obtained from reference 9 was considered. Indeed, the first paragraph of Results (from lane 159) does not contribute anything different from the reference 9. The 112 (111) genes are already in reference 9 predicted target genes for miR-1253 within the IPA gene sets and differentially-expressed in PBMCs. Even WAVE2 appeared in that reference with the name WASF2, the correct name, by the way. The new contributions of this manuscript begin in lane 183, with the in vitro experiments.

We understand the confusion behind this and we thank the reviewers for asking for clarification. Indeed, we did perform a reanalysis of our prior hypertension data set [4] to further examine the actin cytoskeleton pathway and in conjunction with our new microarray analysis (for this paper) of miR-1253 over-expression in endothelial cells. We clarified this information in the abstract (line 19), introduction (line 81), methods (line 88) and in the beginning of the results (lines 163-167) to emphasize that this study features a re-analysis of prior microarray data in hypertensive women, as well as a new microarray of miR-1253 target analysis in endothelial cells. In response to your minor comment below, we have changed the name of WAVE2 to the more conventional WASF2 throughout the entire manuscript and supplemental information, as well as within the figures, to address this inconsistency.

In our prior study, while the actin-cytoskeleton pathway was briefly discussed with respect to our findings with RHOA expression, and included in our hypertension-related dataset, we did not perform a detailed analysis of DGE in this pathway until we began the work featured in this paper. In this context, all of our results presented here are new findings, taken in context as a follow-up to our prior paper. We hope the changes we described above have helped clarify this point in the manuscript.

  1. In Material and Methods also, transfection experiments are not clear. Are miR-1253 pre-miRNA (precursors) or mimics (mature) used? Why 50 nM is used? Did you try different concentrations? Are transfections for luciferase assays performed also with Lipofectamine 2000?

In lines 115 of the edited document, we have already mentioned that we used miR-1253 and negative control #1 pre-miR miRNA precursors. We chose 50 nM as a standard transfection concentration for our 6-well and 10 cm plates as an optimal concentration range for the HAECs and HUVECs and is the same concentration featured in many experiments in our prior work [4-6]. HAECs and HUVECs can often exhibit toxicity-related issues when transfected with higher concentrations. For this experiment, we did not try lower concentrations of miR-1253 mimic. We did use Lipofectamine 2000 for our luciferase assays and we have updated our methods to reflect this (lines 124).  

  1. Supplementary File 1 is very confusing. Not all the lists of genes are in alphabetical order. Please, homogenize. Also, please review gene numbers in sets because, for example, the number of predicted targets that are also found in the list of 1,266 genes related to hypertension and inflammation is 111, not 112 (lane 167). In addition, these 1,266 genes are not only those related to hypertension and inflammation but those included in all IPA sets considered.

We have restructured Supplementary File 1 so that is easier to follow and we have alphabetized each list. We have separated each gene list resulting from Figure 1 into a separate tab and labeled each column to provide more clarity as the gene lists of potential miR-1253 targets was/is pared down. We thank the reviewer for pointing out our mistake on the numbers of genes in the text and within the supplemental files. This was pointed out by another reviewer who had additional comments on our target analysis. We have since re-analyzed part of our dataset and identified 117, not 111 (new lines 183=190) that overlap with our gene list and differentially-expressed in hypertensive women. Below is our response to this critique which will explain this. As well, we have updated in the text (lines 175-176) to stress that our hypertension gene list of 1,266 genes is an IPA- and manually-curated list from our prior study.

‘When reconfirming this during our revisions, we realized we made a mistake in citing the wrong paper and DIANA algorithm for our miR-1253 analysis. The DIANA database provides several different suites of algorithms for microRNA predication analysis. We mistakenly cited the DIANA-Tarbase algorithm, but our gene lists and binding site predictions had been generated using the DIANA-microT algorithm (both are found in the DIANA toolkit). We have updated our methods section, results, and references list to cite the correct algorithm (lines 109).

In addition, we decided to re-run and confirm our DIANA-microT miR-1253 predictions, as this analysis was first performed some time ago. Our re-analysis identified several hundred more potential targets for miR-1253. Following through our gene lists comparisons in the Venn diagram workflow of Figure 1, the number of potential targets in hypertension-related pathways ultimately increased from 111 to 117 (Figure 1C). We have updated Figure 1 and the text, as well as the supplemental files, to reflect this update. Importantly, when comparing the final predicted 117 genes (originally 111) to those targets down-regulated in our miR-1253 over-expression in HAECs, we identify the same (and only the same) 23 mRNAs as we did in our initial target analysis for miR-1253 (Table 1). We felt it prudent to update these lists but this does not change the identification of WAVE2 as a predicted target. This is also in response to another reviewer’s critiques of the supplemental file and gene lists.  

  1. In lane 201 it is stated that 75 genes are in the Actin Cytoskeleton signaling pathway but only 54 genes are shown in Supp File 1. Even, neither RHOA nor WAVE2 are in this list of genes. Maybe the correct list is the one in reference 9, with 84 genes, which includes WAVE2? In this column, what are the genes in 842-862 rows? It is very confusing.

We greatly appreciate the opportunity to clarify this. The correct list is now presented in the last tab of the Supplementary File 1 entitled “HAEC Target and Pathway Analysis” and in a separate tab.  This list has all 75 genes, including RHOA and WASF2. The confusing rows in this column have been removed, as we weren’t sure why that was there and we suspect we may have attached the wrong supplementary draft file. This has now been rectified.

  1. In lane 226, authors compare the 747 repressed mRNAs with the hypertension gene list. Why do they not with the Actin Cytoskeleton list? Considering that WAVE2 is in this last list.

The 747 mRNAs identified in this comparison are a result of comparing the repressed transcripts in HAECs due to miR-1253 over-expression and the hypertension gene list of 1,266 genes. We then compared these 747 mRNAs with the list of 117 predicted targets that are differentially-expressed in hypertensive women (Figure 1D). This results in the 23 genes listed in Table 1 that overlap between these two lists, of which WAVE2 is the only member of the actin-cytoskeleton significant in all cases. We have updated lines 253-268 to clarify this with :

“In order to determine whether miR-1253 might play a role in the differential-expression of genes within the actin cytoskeleton signaling pathway, we compared the list of mRNAs significantly down-regulated in HAECs via over-expression of miR-1253 mimic against the list of 117 predicted miR-1253 targets which were differentially-expressed in hypertensive women (Figure 1C). There were 747 mRNAs significantly repressed >1.5-fold compared with the scrambled negative control (P<0.05; FDR <0.20; n=5; Supp. File 1). Of these 747, 23 mRNAs overlapped with our list of 117 predicted targets and which were differentially-expressed in hypertensive women (Table 1). When we compared this list with the 75 genes in the actin cytoskeleton pathway, one of these genes, WASP family Verprolin-homologous protein 2 WASF2 (also known as WAVE2) is differentially-expressed between AAHT and WHT women (Figure 2, circled in purple) and regulates actin cytoskeleton branching [7, 8]. miR-1253 is also predicted to target two other genes in the actin cytoskeleton pathway, Filamin A, Alpha (FLNA) and Ras Homolog A (RHOA), however, neither of these two mRNAs were significantly down-regulated by miR-1253 in our screen. Therefore, we focused on WASF2 as a potential target of miR-1253.”

We approached it this way to filter out genes that weren’t significantly different in hypertensive women. It is possible we have filtered out some true positives in this approach, but we were able to use to isolate and validate WASF2.

  1. In lane 335, “AA women have higher levels of FLNA compared with white women” is true just in case of hypertensive women. In any case, FLNA has the same sign (negative) than WAVE2, for example, and WAVE2 is described here to have lower levels in AAHT. It is really confusing.

Thank you for identifying this error. We meant to say ‘lower’, not higher, and this has been changed in the discussion (line 387). We also added this line to clarify our thinking on this issue (line 387-390):

“In our analysis, we found that AA women have lower levels of FLNA compared with white women which may suggest that FLNA and its expression in specific contexts is relevant in hypertension etiology”

  1. The first paragraph of the discussion (from lane 312) includes basically results from the previous work of the authors but it is presented as if it had been obtained in the present manuscript. In addition, I cannot find sense to this sentence in lane 318: “further evidence that DGE is associated with individual gene expression levels in individuals with high blood pressure”.

We have rewritten the opening sentences of our discussion to remedy this and clarify and it now says (lines 361-364):

“Together, reanalysis of unexplored pathways within our prior data examining DGE patterns in AA and white women with hypertension indicates that a large number of genes within the actin cytoskeleton signaling pathway are differentially-expressed between AA and white hypertensive women.”

We have also reworded and edited line 318 (new line 368) to clarify our meaning with the following sentence:

“Previously, we found similar patterns in additional pathways related to hypertension [4, 5] and this study provides further evidence that DGE is associated with individuals with high blood pressure.”

  1. The second part of the statement (in lane 322) “… validate that miR-1253 can bind and regulate WAVE2 expression in endothelial cells and influence actin cytoskeletal dynamic” cannot be deduced from the results presented here.

We have reworded this sentence to narrow the focus of this statement to reflect the findings presented here. We believe that with the use of the luciferase reporter assay, mRNA and protein expression differences in the presence of miR-1253 mimic, and the increase in cell area and lamellipodia formation in these cells, we have provided sufficient evidence that miR-1253 is capable of regulating WASF2 expression.

“Our gene expression analysis in PBMCs led us to identify that miR-1253 can regulate the WASF2 3’ UTR, repress WASF2 mRNA and protein expression in endothelial cells, and influence actin cytoskeletal dynamics with respect to causing increased lamellipodia formation in endothelial cells(Figures 4-5).”

  1. In lane 391, the use of bioinformatic analysis to identify and validate novel miRNA regulators for members of that pathway is not identified in the present manuscript and, the fact that WAVE2 could be validated for the authors as a miR-1253 target is not guarantee to validate other future bioinformatic findings.

We have removed this sentence from our manuscript as we agree after reviewing these comments, this is not a guarantee that others can be validated in this way.  We appreciate this input and recommendation.

Minor comments

  1. Gene names are mixed with protein names. The attempt to put gene names in italics is not useful if the official gene name is not used. WAVE2 should be WASF2.

We have gone through the manuscript and all supporting files to change the name of WAVE2 to WASF2, including within the title and abstract, as well as the diagram of Figure 2 and Supplemental Figure 1.

  1. It is impossible to see Figure 2. Please, enhance it. But also labels in Figure 4 and Figure 5B and C.

We thank the reviewer for these comments. We have increased the fonts of the figures and made changes to Figures 1-5 per yours and other reviewer comments as well. We have also increased the font size to make them more legible. The font of the words on all of the graph axes has been increased for visibility.

We have also adjusted Figure 2 so that it is larger in response to your and another reviewer’s concerns. As the comparison between AAHT vs WHT is the most important, we have removed Part 2A in this figure and moved it to Supplementary Figure 1A. This leaves Figure 2B as the sole diagram of the new Figure 2. We have also updated the Figure legend and corresponding results sections to reflect his modification.

  1. What does it mean “F-actin” in the list of genes? To my knowledge, F-actin is a polymer of actin, not a unique protein encoded by a gene. ACTG1 codes for gamma-actin.

The diagram in Figure 2 shows F-actin in lieu of ACTG1 as this pathway representation was curated in Ingenuity Pathway Analysis and IPA often will compress several genes together into one representative figure in some cases. The heat map data generated for the F-actin symbol is reflective of gene expression of ACTG1 overlaid from our microarray and we have both names listed in Supp. Table 1 to reflect this and provide consistency. We have also updated the results section (lines 235) to clarify on this issue.

  1. In lane 327, “plaques” instead of “plagues”.

Thank you for catching this. It has been changed.

  1. In lane 387, “It is unknown…”.

Thank you for catching this. It has been changed.

References:

  1. Leung, A. K., and P. A. Sharp. "Microrna Functions in Stress Responses." Mol Cell 40, no. 2 (2010): 205-15.
  2. Mukherji, S., M. S. Ebert, G. X. Zheng, J. S. Tsang, P. A. Sharp, and A. van Oudenaarden. "Micrornas Can Generate Thresholds in Target Gene Expression." Nat Genet 43, no. 9 (2011): 854-9.
  3. Schmiedel, J. M., S. L. Klemm, Y. Zheng, A. Sahay, N. Bluthgen, D. S. Marks, and A. van Oudenaarden. "Gene Expression. Microrna Control of Protein Expression Noise." Science 348, no. 6230 (2015): 128-32.
  4. Dluzen, D. F., N. Noren Hooten, Y. Zhang, Y. Kim, F. E. Glover, S. M. Tajuddin, K. D. Jacob, A. B. Zonderman, and M. K. Evans. "Racial Differences in Microrna and Gene Expression in Hypertensive Women." Sci Rep 6 (2016): 35815.
  5. Dluzen, D. F., Y. Kim, P. Bastian, Y. Zhang, E. Lehrmann, K. G. Becker, N. Noren Hooten, and M. K. Evans. "Micrornas Modulate Oxidative Stress in Hypertension through Parp-1 Regulation." Oxid Med Cell Longev 2017 (2017): 3984280.
  6. Kim, Y., N. Noren Hooten, D. F. Dluzen, J. L. Martindale, M. Gorospe, and M. K. Evans. "Posttranscriptional Regulation of the Inflammatory Marker C-Reactive Protein by the Rna-Binding Protein Hur and Microrna 637." Mol Cell Biol 35, no. 24 (2015): 4212-21.
  7. Beli, P., D. Mascheroni, D. Xu, and M. Innocenti. "Wave and Arp2/3 Jointly Inhibit Filopodium Formation by Entering into a Complex with Mdia2." Nat Cell Biol 10, no. 7 (2008): 849-57.
  8. Krause, M., and A. Gautreau. "Steering Cell Migration: Lamellipodium Dynamics and the Regulation of Directional Persistence." Nat Rev Mol Cell Biol 15, no. 9 (2014): 577-90.

Reviewer 2 Report

In the manuscript submitted by Arkorful et al, the authors aimed to assess the role of the miR-1253 in the regulation of the WAVE gene as potential molecular mechanisms that could be involved in the higher rate of hypertension in African Americans. The study is well done, manuscript well written and provide novel very important results. There are few points that authors could clarify:

  • How the regulation of the actin cytoskeleton organization is linked to hypertension? The authors did provide relevant information regarding the role of this pathway in atherosclerosis and endothelial function but what about the link with hypertension?
  • The authors could provide a summary figure presenting their findings and links between miRNA and genes of interest
  • Are there publications on known ways to decrease the expression of miR-1253, like drugs or nutrients/diets?

Author Response

Response to Review 2:

Dear Reviewer,

The authors thank you for providing a thorough and critical review our manuscript. We appreciate the opportunity to improve our work. Below, we have provided our responses and detailed the changes we have made to the manuscript to address these concerns. We thank the reviewer again for their time and hope this provides clarification.

Regards,

Doug Dluzen

In the manuscript submitted by Arkorful et al, the authors aimed to assess the role of the miR-1253 in the regulation of the WAVE gene as potential molecular mechanisms that could be involved in the higher rate of hypertension in African Americans. The study is well done, manuscript well written and provide novel very important results. There are few points that authors could clarify:

  1. How the regulation of the actin cytoskeleton organization is linked to hypertension? The authors did provide relevant information regarding the role of this pathway in atherosclerosis and endothelial function but what about the link with hypertension?

We thank the reviewer for this comment. We have included a new paragraph in the Discussion section to address this and provide some context for the role of the actin cytoskeleton pathway in hypertension, particularly within PBMCs. Specifically, the last few sentences in this paragraph (421-435).

“Modulation of WASF2 expression by miR-1253 in circulating PBMCs may contribute towards hypertension-related changes in membrane physiology and morphology and downstream complications, such as atherosclerosis. Follow-up studies will be necessary to ascertain this. Recent findings suggest that B cells [1] and several T cell subtypes can influence Angiotensin II (AngII)-signaling and downstream cytokine release and inflammatory response in hypertensives [1-4]. M-positive monocytes mediate AngII-induced hypertension and promote downstream vascular dysfunction in response to elevated blood pressure [5].While there have been few studies directly investigating the role of the actin cytoskeleton structure in these cells as a causative mechanism of hypertension, several members of this pathway, including RHOA and ROCK, have identified roles in T and other immune cell cytoskeletal structure [6, 7]. This includes regulatory roles in lamellipodial function required for trans-endothelial migration during inflammatory-response [6, 8]. Our findings indicate that WASF2 may also be a player in immune cell cytoskeletal dynamics and future studies will help elucidate this contribution to any additional hypertension-related mechanisms.”

  1. The authors could provide a summary figure presenting their findings and links between miRNA and genes of interest

We thank you for this consideration and thought heavily on how we might approach this. However, we ultimately felt that it wasn’t necessary only due to the already heavy reliance on several figures and extensive supplemental datasets, with the pathways visualized.

  1. Are there publications on known ways to decrease the expression of miR-1253, like drugs or nutrients/diets?

We agree this is a very interesting and important point. However, we were unable to find any literature related to this with respect to diet or nutrition. This is likely due to the small number of publications on this microRNA. We did find that a microRNA sponge can repressed endogenous miR-1253 in pancreatic cells, causing cellular growth. We have updated our discussion about what is known about miR-1253 to reflect this (lines 439-440).

  1. Chan, C. T., C. G. Sobey, M. Lieu, D. Ferens, M. M. Kett, H. Diep, H. A. Kim, S. M. Krishnan, C. V. Lewis, E. Salimova, P. Tipping, A. Vinh, C. S. Samuel, K. Peter, T. J. Guzik, T. S. Kyaw, B. H. Toh, A. Bobik, and G. R. Drummond. "Obligatory Role for B Cells in the Development of Angiotensin Ii-Dependent Hypertension." Hypertension 66, no. 5 (2015): 1023-33.
  2. Caillon, A., M. O. R. Mian, J. C. Fraulob-Aquino, K. G. Huo, T. Barhoumi, S. Ouerd, P. R. Sinnaeve, P. Paradis, and E. L. Schiffrin. "Gammadelta T Cells Mediate Angiotensin Ii-Induced Hypertension and Vascular Injury." Circulation 135, no. 22 (2017): 2155-62.
  3. Ni, X., A. Wang, L. Zhang, L. Y. Shan, H. C. Zhang, L. Li, J. Q. Si, J. Luo, X. Z. Li, and K. T. Ma. "Up-Regulation of Gap Junction in Peripheral Blood T Lymphocytes Contributes to the Inflammatory Response in Essential Hypertension." PLoS One 12, no. 9 (2017): e0184773.
  4. Drummond, G. R., A. Vinh, T. J. Guzik, and C. G. Sobey. "Immune Mechanisms of Hypertension." Nat Rev Immunol 19, no. 8 (2019): 517-32.
  5. Wenzel, P., M. Knorr, S. Kossmann, J. Stratmann, M. Hausding, S. Schuhmacher, S. H. Karbach, M. Schwenk, N. Yogev, E. Schulz, M. Oelze, S. Grabbe, H. Jonuleit, C. Becker, A. Daiber, A. Waisman, and T. Munzel. "Lysozyme M-Positive Monocytes Mediate Angiotensin Ii-Induced Arterial Hypertension and Vascular Dysfunction." Circulation 124, no. 12 (2011): 1370-81.
  6. Heasman, S. J., and A. J. Ridley. "Multiple Roles for Rhoa During T Cell Transendothelial Migration." Small GTPases 1, no. 3 (2010): 174-79.
  7. Ricker, E., L. Chowdhury, W. Yi, and A. B. Pernis. "The Rhoa-Rock Pathway in the Regulation of T and B Cell Responses." F1000Res 5 (2016).
  8. Konigs, V., R. Jennings, T. Vogl, M. Horsthemke, A. C. Bachg, Y. Xu, K. Grobe, C. Brakebusch, A. Schwab, M. Bahler, U. G. Knaus, and P. J. Hanley. "Mouse Macrophages Completely Lacking Rho Subfamily Gtpases (Rhoa, Rhob, and Rhoc) Have Severe Lamellipodial Retraction Defects, but Robust Chemotactic Navigation and Altered Motility." J Biol Chem 289, no. 44 (2014): 30772-84.

Reviewer 3 Report

In this manuscript Arkorful et al., have examined in further detail their previously published data on differential gene expression in African American (AA) women with hypertension in comparison with white hypertensive women. They identify a role for the hypertension-related miR-1253 in the regulation of WAVE2 gene expression. The manuscript is interesting, and is overall properly performed and well written. However, before publication can be recommended, it would be important to clarify or correct some of the analyses performed and include some additional experiments. Here are the specific points that need to be addressed:

  • The results described in Figure 1 are very confusing and it is hard to follow the line of reasoning in this figure. What is the rationale for the analysis shown in figure 1A,B,C if those data will not be used for the remainder of the manuscript? Did the authors include both the upregulated and downregulated set of genes in the same analysis? Does this type of analysis consider whether genes show an increased or decreased expression? The results from this figure should be rewritten in a clearer way.
  • No reason is provided in the manuscript for focusing on the actin cytoskeleton signaling pathway. Are genes from the actin cytoskeleton signaling pathway present in any of the enriched pathways shown in Figure 1D? Which genes? A deeper analysis of this pathway in the microarray performed in the present manuscript upon miR-1253 mimic overexpression would make the reader understand better the focus of the authors on this signaling pathway.
  • TargetScan predicted that miR-1253 binds to the WAVE2 3’ UTR at nucleotides 3,734-3,756. How were the two additional positions at nucleotides 1,617 and 1,775 at the 3’UTR identified? Please specify. Would a different search tool for miRNA targets, such as the DIANA-TARBASE used in Figure 1, find these other sites not found by TargetScan?
  • In Figure 5, miR-1253 mimics are transfected to assess the effect of this miRNA in actin cytoskeletal structures. The phenotypes obtained upon miR-1253 mimic overexpression should be compared with the effect of silencing the expression of WAVE2 to assess how much the observed phenotype is the result of WAVE2 downregulation. Alternatively, the authors could perform a rescue experiment overexpressing WAVE2 in miRNA expressing cells.
  • Also in Figure 5, please indicate in the IF figures, or using new IF figures, examples of the lamellipodia and filopodia structures that are scored in the analysis. How many cells were scored for these analyses? Please, include the standard deviation for the scramble control. Moreover, is the concentration of actin-rich membrane-ruffling at the edges of cells enough to define these types of structures? If so, could the authors indicate any references for this type of scoring? Otherwise, this reviewer would suggest not being so definitive in the conclusions that are drawn.
  • In the Discussion section, it would be useful to speculate on why the expression of miR-1253 might be decreased in hypertensive AA women.
  • In Figure 4 standard deviations for the Scramble sample are missing. Please correct.
  • Figure 2 is too small to see anything Moreover, showing Figure 2A seems unnecessary, considering that only PAK is differentially expressed

Author Response

Response to Review 3:

Dear Reviewer,

The authors thank you for providing a thorough and critical review our manuscript. We appreciate the opportunity to improve our work. Below, we have provided our responses and detailed the changes we have made to the manuscript to address these concerns. We thank the reviewer again for their time and hope this provides clarification.

Regards,

Doug Dluzen

  1. The results described in Figure 1 are very confusing and it is hard to follow the line of reasoning in this figure. What is the rationale for the analysis shown in figure 1A,B,C if those data will not be used for the remainder of the manuscript? Did the authors include both the upregulated and downregulated set of genes in the same analysis? Does this type of analysis consider whether genes show an increased or decreased expression? The results from this figure should be rewritten in a clearer way.

We thank the reviewers for raising this issue. Our rationale for this approach was to begin to walk the reader through how we pared down thousands of predicted miR-1253 targets to our focus on WAVE2.

We have reworded the results section featuring Figures A-C (lines 163-190) in order to clarify our reasoning for this analysis, as well as fixed typographic errors in the Figure 1 Venn diagrams to accurately reflect the correct number of genes in each comparison. We hope these changes provide more clarify on how we approached paring down the predicted mRNA gene lists. We also added a statement clarifying that in order to be as comprehensive as possible in paring down the thousands of predicted genes for miR-1253, we did not consider directionality of the target genes from our prior analysis of gene expression in hypertensive women. We have also updated and clarified the naming of the tabs within the Supplementary File to reflect this re-structuring as well as to reflect an additional reviewer’s comments. As well, we have updated Figure 1 due to minor changes in our analysis (more detail provided in response to Comment #3).

  1. No reason is provided in the manuscript for focusing on the actin cytoskeleton signaling pathway. Are genes from the actin cytoskeleton signaling pathway present in any of the enriched pathways shown in Figure 1D? Which genes? A deeper analysis of this pathway in the microarray performed in the present manuscript upon miR-1253 mimic overexpression would make the reader understand better the focus of the authors on this signaling pathway.

“In our results section, we have updated our reasoning for choosing this pathway and clarified this with the following edit (lines 215-226):

We next examined DGE in the actin cytoskeleton signaling pathway in our hypertension cohort by reanalyzing our previous microarray dataset GSE75672 from [1] (gene list in Supp. File 1). We chose this pathway for several reasons. First, given the role of actin cytoskeletal remodeling and signaling in hypertension and endothelial function [2-4]. Second, given the importance of this pathway in the Cardiovascular Diseases category identified in our IPA analysis (Figure 1D), in which 16 genes in the actin cytoskeleton pathway are found in the Cardiovascular Disease molecules list in IPA (see Supp. File 1). Third, our prior analysis that RHOA, a member of the actin cytoskeleton pathway and associated with hypertension etiology in endothelial and vascular smooth muscle cells, is differentially expressed in PBMCs between AA and white hypertensive women. And fourth, the fact that this particular pathway in PBMCs is relatively unexplored with respect to hypertension.”

This list of 16 genes, which includes WASF2, RHOA, and others in the actin cytoskeleton pathway, are listed in Supplementary File 1 in the Cardiovascular Disease Pathway Tab and includes:

Overlap

ABI2

ARHGEF1

CD14

CDC42

CRK

DOCK1

F2

F2R

FLNA

FN1

GRB2

GSN

NWASP

RHOA

VCL

WASF2 (WAVE2)

  1. TargetScan predicted that miR-1253 binds to the WAVE2 3’ UTR at nucleotides 3,734-3,756. How were the two additional positions at nucleotides 1,617 and 1,775 at the 3’UTR identified? Please specify. Would a different search tool for miRNA targets, such as the DIANA-TARBASE used in Figure 1, find these other sites not found by TargetScan?

There can be differences in binding site predictions based on the weight of the criteria used for each algorithm and the filters used. For example, conservation of seed sequences across animal kingdoms is favored in programs like TargetScan, but that can be to the detriment of identifying predicted targets if the transcript or miRNA being searched is only found in humans.

The two additional positions in this paper were identified using the DIANA suite of tools, specifically DIANA-microT. The DIANA-microT algorithm identifies Binding Sites #1, #2, and #3, and the identification of binding site #3 is also the binding site identified by TargetScan. We have updated our results text to clarify this overlap (lines 278-283) as well as updated Figure 3 to provide the predicted schematics for all three binding sites.

When reconfirming this during our revisions, we realized we made a mistake in citing the wrong paper and DIANA algorithm for our miR-1253 analysis. The DIANA database provides several different suites of algorithms for microRNA predication analysis. We mistakenly cited the DIANA-Tarbase algorithm, but our gene lists and binding site predictions had been generated using the DIANA-microT algorithm (both are found in the DIANA toolkit). We have updated our methods section and references list to cite the correct algorithm (line 109).

In addition, we decided to re-run and confirm our DIANA-microT miR-1253 predictions, as this analysis was first performed some time ago. Our re-analysis identified several hundred more potential targets for miR-1253, likely due to updates in software. Following through with our gene lists comparisons in the Venn diagram workflow of Figure 1, the number of potential targets in hypertension-related pathways ultimately increased from 111 to 117 (Figure 1C). We have updated Figure 1 and the text, as well as the supplemental files, to reflect this update. Importantly, when comparing the final predicted 117 genes (originally 111) to those targets down-regulated in our miR-1253 over-expression in HAECs, we identify the same (and only the same) 23 mRNAs as we did in our initial target analysis for miR-1253 (Table 1). We felt it prudent to update these lists but this does not change the identification of WASF2 (WAVE2) as a predicted target. This is also in response to another reviewer’s critiques of the supplemental file and gene lists.  

  1. In Figure 5, miR-1253 mimics are transfected to assess the effect of this miRNA in actin cytoskeletal structures. The phenotypes obtained upon miR-1253 mimic overexpression should be compared with the effect of silencing the expression of WAVE2 to assess how much the observed phenotype is the result of WAVE2 downregulation. Alternatively, the authors could perform a rescue experiment overexpressing WAVE2 in miRNA expressing cells.

We thank the reviewer for this recommendation and we agree in part about these experiments. WAVE2 is an important regulating switch between actin branching and influencing the formation of cellular filopodia and lamellipodia [5-9]. However, microRNA regulation of most genes is often a modulator of gene expression levels and protein translation, not a switch to turn on and off gene expression like silencing can be. By silencing the expression of WASF2 (WAVE2), we may not be representative of that modulation and may see phenotypes that wouldn’t necessarily represent miR-1253 modulation. We do acknowledge that over-expression of miR-1253 is only an approximation of endogenous miR-1253 in endothelial or immune cells.

We agree that a WASF2-rescue of cells transfected with miR-1253 is a worthwhile experiment. However, currently the laboratories of Drs. Dluzen and his colleagues at the NIH are shutdown due to the COVID-19 pandemic. We are not able to perform these experiments in a timely fashion. We have included in the discussion a list of follow-up experiments that would be required to confirm this and mention these experiments as possibilities (lines 432-434; 461-463). We ask the reviewer to consider this in their review of our response.

  1. Also in Figure 5, please indicate in the IF figures, or using new IF figures, examples of the lamellipodia and filopodia structures that are scored in the analysis. How many cells were scored for these analyses? Please, include the standard deviation for the scramble control. Moreover, is the concentration of actin-rich membrane-ruffling at the edges of cells enough to define these types of structures? If so, could the authors indicate any references for this type of scoring? Otherwise, this reviewer would suggest not being so definitive in the conclusions that are drawn.

As suggested by the reviewer, we have indicated on the immunofluorescence figures lamellipodia and filopodia structures. We have also provided the S.D. for the scrambled control bar across all three experiments and indicated this in the figure legend, along with the total cell count ranges for the scrambled control and mimic columns (lines 356-358). For our first experiment, we scored 89 controls cells and 46 mimics, for the second 28 control cells and 25 mimics, and for the third 61 control cells and 31 mimics. Because control cells were smaller, we were able to count more per image.

Lamellipodia and filopodia are classically defined as we referenced in the text as “Protrusive actin-containing structures such as lamellipodia or filopodia are formed at the leading edge of cells. Lamellipodia form larger actin-containing ruffles while filopodia are characterized by actin-containing finger-like extensions from the cell.” (lines 333-336). Decades of research in this field has shown that these structures are regulated by different proteins that drive actin polymerization and bundling in these protrusions. These structures are most readily identified by staining the cells with actin.  For increased clarity, we have added this sentence to the results “

“These structures can be identified by immunofluorescence staining of cells for actin” on line 336 and also included relevant reviews on lamellipodia and filopodia as references.

In the Materials and Methods, we referenced that we scored the cells according to Beli et al., Nat. Cell Biol. 2008. However, we realize that this reference may have been missed since we included it at the end of the section.  Therefore, we have added it in earlier in the section along with other references that score actin-rich structures such as lamellipodia and filopodia using this technique, which has been a conventional method in the field. [5, 10-12]

  1. In the Discussion section, it would be useful to speculate on why the expression of miR-1253 might be decreased in hypertensive AA women.

We have added the following sentences in the discussion (lines 458-463) to address this comment.

“It is interesting to speculate how AA women with hypertension may have lost expression of miR-1253 in PMBCs. Considering the miR-1253 loci is sensitive to hypermethylation in cancer [13], this could be a potential mechanism repressing miR-1253 expression levels. As well, ancestral polymorphisms can function as expression quantitative trait loci (eQTL) in macrophages and immune cells to influence differential gene expression and immune fuction[14]. It is possible miR-1253 expression is regulated by eQTLs in AA hypertensives and this will need to be explored further, as this would also provide a more thorough understanding if miR-1253 expression changes before, during, or after the development of essential hypertension.”

  1. In Figure 4 standard deviations for the Scramble sample are missing. Please correct.

We have provided the standard deviation for all scrambled controls in Figure 4 and updated the Figure.

  1. Figure 2 is too small to see anything Moreover, showing Figure 2A seems unnecessary, considering that only PAK is differentially expressed

We thank the reviewer for bringing this to our attention. Part of this is due to the formatting of manuscripts in the MDPI system and we didn’t realize this at the time of submission. We have adjusted Figure 2 so that it is larger. As the comparison between AAHT vs WHT is the most important, we have removed Part 2A in this figure and moved it Supplementary Figure 1A. This leaves Figure 2B as the sole diagram of the new Figure 2. We have also updated the Figure legend and corresponding results sections to reflect his modification.

References:

  1. Dluzen, D. F., N. Noren Hooten, Y. Zhang, Y. Kim, F. E. Glover, S. M. Tajuddin, K. D. Jacob, A. B. Zonderman, and M. K. Evans. "Racial Differences in Microrna and Gene Expression in Hypertensive Women." Sci Rep 6 (2016): 35815.
  2. Davies, P. F. "Hemodynamic Shear Stress and the Endothelium in Cardiovascular Pathophysiology." Nat Clin Pract Cardiovasc Med 6, no. 1 (2009): 16-26.
  3. Iskratsch, T., H. Wolfenson, and M. P. Sheetz. "Appreciating Force and Shape-the Rise of Mechanotransduction in Cell Biology." Nat Rev Mol Cell Biol 15, no. 12 (2014): 825-33.
  4. Spindler, V., N. Schlegel, and J. Waschke. "Role of Gtpases in Control of Microvascular Permeability." Cardiovasc Res 87, no. 2 (2010): 243-53.
  5. Beli, P., D. Mascheroni, D. Xu, and M. Innocenti. "Wave and Arp2/3 Jointly Inhibit Filopodium Formation by Entering into a Complex with Mdia2." Nat Cell Biol 10, no. 7 (2008): 849-57.
  6. Innocenti, M., S. Gerboth, K. Rottner, F. P. Lai, M. Hertzog, T. E. Stradal, E. Frittoli, D. Didry, S. Polo, A. Disanza, S. Benesch, P. P. Di Fiore, M. F. Carlier, and G. Scita. "Abi1 Regulates the Activity of N-Wasp and Wave in Distinct Actin-Based Processes." Nat Cell Biol 7, no. 10 (2005): 969-76.
  7. Krause, M., and A. Gautreau. "Steering Cell Migration: Lamellipodium Dynamics and the Regulation of Directional Persistence." Nat Rev Mol Cell Biol 15, no. 9 (2014): 577-90.
  8. Rotty, J. D., C. Wu, and J. E. Bear. "New Insights into the Regulation and Cellular Functions of the Arp2/3 Complex." Nat Rev Mol Cell Biol 14, no. 1 (2013): 7-12.
  9. Suarez, C., and D. R. Kovar. "Internetwork Competition for Monomers Governs Actin Cytoskeleton Organization." Nat Rev Mol Cell Biol 17, no. 12 (2016): 799-810.
  10. Isogai, T., R. van der Kammen, D. Leyton-Puig, K. M. Kedziora, K. Jalink, and M. Innocenti. "Initiation of Lamellipodia and Ruffles Involves Cooperation between Mdia1 and the Arp2/3 Complex." J Cell Sci 128, no. 20 (2015): 3796-810.
  11. Schaks, M., H. Doring, F. Kage, A. Steffen, T. Klunemann, W. Blankenfeldt, T. Stradal, and K. Rottner. "Rhog and Cdc42 Can Contribute to Rac-Dependent Lamellipodia Formation through Wave Regulatory Complex-Binding." Small GTPases (2019): 1-11.
  12. Steffen, A., M. Ladwein, G. A. Dimchev, A. Hein, L. Schwenkmezger, S. Arens, K. I. Ladwein, J. Margit Holleboom, F. Schur, J. Victor Small, J. Schwarz, R. Gerhard, J. Faix, T. E. Stradal, C. Brakebusch, and K. Rottner. "Rac Function Is Crucial for Cell Migration but Is Not Required for Spreading and Focal Adhesion Formation." J Cell Sci 126, no. Pt 20 (2013): 4572-88.
  13. Kanchan, R. K., N. Perumal, P. Atri, R. Chirravuri Venkata, I. Thapa, D. L. Klinkebiel, A. M. Donson, D. Perry, M. Punsoni, G. A. Talmon, D. W. Coulter, D. R. Boue, M. Snuderl, M. W. Nasser, S. K. Batra, R. Vibhakar, and S. Mahapatra. "Mir-1253 Exerts Tumor-Suppressive Effects in Medulloblastoma Via Inhibition of Cdk6 and Cd276 (B7-H3)." Brain Pathol (2020).
  14. Nedelec, Y., J. Sanz, G. Baharian, Z. A. Szpiech, A. Pacis, A. Dumaine, J. C. Grenier, A. Freiman, A. J. Sams, S. Hebert, A. Page Sabourin, F. Luca, R. Blekhman, R. D. Hernandez, R. Pique-Regi, J. Tung, V. Yotova, and L. B. Barreiro. "Genetic Ancestry and Natural Selection Drive Population Differences in Immune Responses to Pathogens." Cell 167, no. 3 (2016): 657-69 e21.

Reviewer 4 Report

General remarks:

The manuscript under review is in fact a sequel/extension of an earlier paper by the same group (Dluzen et al. Sci.Rep. 2016). The same groups of women are researched with largely the same technology, this time focussed on miR-1253 and its target WAVE2. In my opinion, for whatever it is worth, this type of research is important (but limited). Too often are differences in gender or race ignored, although a considerable body of research indicates that the biology of a number of pathologies differs for the different groups and that taking this in account could improve the outcome of medical treatment.

In hypertension two cell types are major players: endothelial and smooth muscle cells. A relevant characteristic of both cell types is the diversity within these cells. Thus, HUVEC cells are different from aorta endothelial cells, which differ from the endothelial cells in the small artheries that play an important role in hypertension.

In this study the investigators start with a first screen in silico on mRNA that are possible targets for miR-1253. In a second screen they analyse in genes selected in the first screen differential expression in PBMCs of the test groups. Why PBMC? The literature shows no solid evidence that PBMCs are causal in hypertension. Changes in PBMCs of hypertensive patients can be considered to be the consequence of hypertension. Subsequently, in a third screen, the 840 selected genes are screened on known involvement in hypertension-related pathways. I am very critical on this approach for two reasons: 1) there is no evidence that the genes selected from de PBMC screen have any causal or direct involvement in the occurrence or maintenance of hypertension; 2) The set of genes affected by miR1253 in endothelial cells (and smooth muscle cells!!) may be incomplete. Important genes/mRNAs may be missed (90% of hypertension related genes is out).

The investigators elaborate on the combi miR1253 and WAVE2. That part of the study well done and provides some interesting new data. The link with the lamellipodia of the endothelial cells is relevant because the endothelium is a dynamic layer. However, if there is one cell type in which the actin cytoskeleton plays a key-role is it is in the smooth muscle cells. More so, since, in the discussion, the investigators refer to phenomena as arterial stiffness. I am left with one question: Is it possible to knock-out WAVE2 and then determine the effect of miR1253.

The set-up and execution of the study lead to a number of associations, suggestions for further studies and needs for validation. Lines 387-394 summarize it very well: There is still a lot of work to do.

Minor remarks:

Fig. 2 (in my copy the figure was too small and hard to read) shows that only one gene was differently expressed between AANT and WNT, whereas more than 30 genes were differentially expressed between AAHT and WHT. In the discussion this huge difference is considered to be the consequence of hypertension. How does this fit with the small numbers of differentially expressed genes in normal and hypertensive individuals of the same race. It does not make sense.

Line 188-190, Cardiovascular Diseases was amongst the “most significantly affected’. What about Reproductive systems disease?

Line 373-375, Hypertension is a complex disease and may have rather different causes. The way the study is set up, makes it unlikely that differential expression of miR1253 is the causal to hypertension.

Author Response

Response to Review 4:

Dear Reviewer,

The authors thank you for providing a thorough and critical review our manuscript. We appreciate the opportunity to improve our work. Below, we have provided our responses and detailed the changes we have made to the manuscript to address these concerns. We thank the reviewer again for their time and hope this provides clarification.

Regards,

Doug Dluzen

  1. In this study, the investigators start with a first screen in silico on mRNA that are possible targets for miR-1253. In a second screen they analyse in genes selected in the first screen differential expression in PBMCs of the test groups. Why PBMC? The literature shows no solid evidence that PBMCs are causal in hypertension. Changes in PBMCs of hypertensive patients can be considered to be the consequence of hypertension.

We thank you for the opportunity to clarify. We choose PBMCs for a few reasons. The first is that when we began our initial study [1] on differential gene expression (DGE) in hypertensive women, PBMCs were a readily- and easily-available biospecimen that gave us a looking glass into gene expression behavior. It is much more difficult to obtain primary endothelial cells or smooth muscle tissue samples in a diverse population. There are many studies that have explored differential gene expression in PBMCs as a marker for gene expression in individuals living in poverty  or for immune function [2-5]

Secondly, while we agree with the reviewer that most analyses of changes in PBMC gene expression patterns do not shed light on whether they are causal of hypertension, we are also examining PBMCs because they are an understudied cell population in essential hypertension and there is growing evidence these cells have a larger role than previously thought in the development and etiology of hypertension. Several studies have identified both contributing and initiating roles in the development of high blood pressure [6-11]. A very thorough review of how subtypes are peripheral blood cells are implicated in high blood pressure initiation and hypertension persistence can be found here [8].

In our Discussion section, we do include added emphasis that our study cannot differentiate between these gene expression differences as a cause or result of hypertension with the following in line 451-453:

“Our study is limited because it is not known whether differential-expression of miR-1253 in AA women with hypertension is a contributing cause or an effect of elevated high blood pressure.”

As well, we have also added the sentence in a new paragraph in the discussion (lines 421-425) to clarify this still has yet to be determined. “Modulation of WASF2 expression by miR-1253 in circulating PBMCs may contribute towards hypertension-related changes in membrane physiology and morphology and downstream complications, such as atherosclerosis. Follow-up studies will be necessary to ascertain this.”

  1. Subsequently, in a third screen, the 840 selected genes are screened on known involvement in hypertension-related pathways. I am very critical on this approach for two reasons: 1) there is no evidence that the genes selected from de PBMC screen have any causal or direct involvement in the occurrence or maintenance of hypertension; 2) The set of genes affected by miR1253 in endothelial cells (and smooth muscle cells!!) may be incomplete. Important genes/mRNAs may be missed (90% of hypertension related genes is out).

The selected genes in the hypertension-related pathways were curated in our initial study on DGE in hypertensive women [1]. The gene sets were collected and compiled from Ingenuity Pathway Analysis [12], which has built a comprehensive disease pathway database built on literature references and (in some cases) bioinformatically-inferred interactions using their algorithms. We built this dataset (detailed in [1]) to include pathways related to hypertension or hypertension-related inflammation based on prior literature. While it’s likely that not every gene in all of the pathways featured in our dataset has a validated role in hypertension, we’re confident the pathways we selected for that prior study and used in this one reflects a majority of major pathways with identified roles in essential hypertension, including genes within the Renin-Angiotensin pathway, nitric oxide signaling pathway, and many others.

Some of these pathways may be less important in PBMCs and were originally identified for their role in hypertension within smooth muscle cells or in endothelial cells. We believe this is a strength of our approach as it is possible some of these genes may also have roles to play in PBMCs but have not yet been identified. We agree, however, this approach may exclude some true positives from our analysis. We are not sure how the comment that ‘90% of hypertension-related genes have been left out’ was calculated, but it’s likely we have missed some relevant genes. Given the role of bioinformatics in this study, it’s likely we could never accomplish encompassing every gene that is possibly hypertension-related. We do believe that while we may have lost some true positives in our approach and in our microarray analysis, it is to the benefit of eliminating as many false positives as possible.

  1. The investigators elaborate on the combi miR1253 and WAVE2. That part of the study well done and provides some interesting new data. The link with the lamellipodia of the endothelial cells is relevant because the endothelium is a dynamic layer. However, if there is one cell type in which the actin cytoskeleton plays a key-role is it is in the smooth muscle cells. More so, since, in the discussion, the investigators refer to phenomena as arterial stiffness. I am left with one question: Is it possible to knock-out WAVE2 and then determine the effect of miR1253.

We appreciate the comments from the reviewer here and this is part of our discussion on future follow-up projects to explore this pathway in greater detail in immune cells and VSMCs. microRNA regulation of most genes is often a modulator of gene expression levels and protein translation, not a switch to turn on and off gene expression like silencing can be. By silencing the expression of WAVE2, however, we may not be representative of that modulation and may see phenotypes that wouldn’t necessarily represent miR-1253 modulation.

  1. The set-up and execution of the study lead to a number of associations, suggestions for further studies and needs for validation. Lines 387-394 summarize it very well: There is still a lot of work to do.

We thank the reviewer for this summary and comment. This study is a beginning in many respects and we believe lays a foundation for follow-up.

Minor remarks:

  1. 2 (in my copy the figure was too small and hard to read) shows that only one gene was differently expressed between AANT and WNT, whereas more than 30 genes were differentially expressed between AAHT and WHT. In the discussion this huge difference is considered to be the consequence of hypertension. How does this fit with the small numbers of differentially expressed genes in normal and hypertensive individuals of the same race. It does not make sense.

We have altered Figure 2 to only emphasize the AAHT vs WHT panel so that it is larger and more clear for the readers and reviewers. The AANT vs WNT panel has also be enlarged and moved to the Supplementary (as per another reviewer comment). We have also re-written a sentence in the first paragraph (see below) of the discussion to reflect your comments that the gene expression differences in AA and white hypertensive women is a consequence of hypertension (line 365-367). We agree that our study does not determine whether these changes are a cause or a consequence of hypertension, just that they exist, and have changed relative to the AANT vs WNT comparison.

“This suggests that the DGE patterns associated with hypertension occur sometime as the disease process begins or during and after sustained exposure to elevated systemic blood pressure levels”.

We also feel that it is important to highlight the normotensive comparison in supplemental, as it shows that gene expression between AA and white women who are normotensive is relatively the same. Few genes have differential expression changes as either hypertension develops or after hypertension has developed. We show that miR-1253 may be a contributing factor this difference and we acknowledge in the discussion (lines 453-463) that there may be other mechanisms that also contribute to these observations, including ancestral alleles and methylation. 

  1. Line 188-190, Cardiovascular Diseases was amongst the “most significantly affected’. What about Reproductive systems disease?

We did not perform an extensive analysis of the gene lists in Reproductive Systems Disease as we wanted to focus our efforts on the Cardiovascular Disease finding. This was predominantly due to the fact that because IPA generates gene sets based on literature-identified associations. We were very interested in CVD-related genes considering how important hypertension is in relation to many CVDs and conditions. We intend to follow-up with the other four Top Diseases and Disorders in the future.

  1. Line 373-375, Hypertension is a complex disease and may have rather different causes. The way the study is set up, makes it unlikely that differential expression of miR1253 is the causal to hypertension.

We have updated the text in the results and discussion to reflect that miR-1253 may be a contributing factor, but not causal, as discussed above.

References:

  1. Dluzen, D. F., N. Noren Hooten, Y. Zhang, Y. Kim, F. E. Glover, S. M. Tajuddin, K. D. Jacob, A. B. Zonderman, and M. K. Evans. "Racial Differences in Microrna and Gene Expression in Hypertensive Women." Sci Rep 6 (2016): 35815.
  2. Cole, S. W., M. J. Shanahan, L. Gaydosh, and K. M. Harris. "Population-Based Rna Profiling in Add Health Finds Social Disparities in Inflammatory and Antiviral Gene Regulation to Emerge by Young Adulthood." Proc Natl Acad Sci U S A 117, no. 9 (2020): 4601-08.
  3. Gaye, A., G. H. Gibbons, C. Barry, R. Quarells, and S. K. Davis. "Influence of Socioeconomic Status on the Whole Blood Transcriptome in African Americans." PLoS One 12, no. 12 (2017): e0187290.
  4. Nedelec, Y., J. Sanz, G. Baharian, Z. A. Szpiech, A. Pacis, A. Dumaine, J. C. Grenier, A. Freiman, A. J. Sams, S. Hebert, A. Page Sabourin, F. Luca, R. Blekhman, R. D. Hernandez, R. Pique-Regi, J. Tung, V. Yotova, and L. B. Barreiro. "Genetic Ancestry and Natural Selection Drive Population Differences in Immune Responses to Pathogens." Cell 167, no. 3 (2016): 657-69 e21.
  5. Shang, L., J. A. Smith, W. Zhao, M. Kho, S. T. Turner, T. H. Mosley, S. L. R. Kardia, and X. Zhou. "Genetic Architecture of Gene Expression in European and African Americans: An Eqtl Mapping Study in Genoa." Am J Hum Genet 106, no. 4 (2020): 496-512.
  6. Caillon, A., M. O. R. Mian, J. C. Fraulob-Aquino, K. G. Huo, T. Barhoumi, S. Ouerd, P. R. Sinnaeve, P. Paradis, and E. L. Schiffrin. "Gammadelta T Cells Mediate Angiotensin Ii-Induced Hypertension and Vascular Injury." Circulation 135, no. 22 (2017): 2155-62.
  7. Chan, C. T., C. G. Sobey, M. Lieu, D. Ferens, M. M. Kett, H. Diep, H. A. Kim, S. M. Krishnan, C. V. Lewis, E. Salimova, P. Tipping, A. Vinh, C. S. Samuel, K. Peter, T. J. Guzik, T. S. Kyaw, B. H. Toh, A. Bobik, and G. R. Drummond. "Obligatory Role for B Cells in the Development of Angiotensin Ii-Dependent Hypertension." Hypertension 66, no. 5 (2015): 1023-33.
  8. Drummond, G. R., A. Vinh, T. J. Guzik, and C. G. Sobey. "Immune Mechanisms of Hypertension." Nat Rev Immunol 19, no. 8 (2019): 517-32.
  9. Ni, X., A. Wang, L. Zhang, L. Y. Shan, H. C. Zhang, L. Li, J. Q. Si, J. Luo, X. Z. Li, and K. T. Ma. "Up-Regulation of Gap Junction in Peripheral Blood T Lymphocytes Contributes to the Inflammatory Response in Essential Hypertension." PLoS One 12, no. 9 (2017): e0184773.
  10. Siedlinski, M., E. Jozefczuk, X. Xu, A. Teumer, E. Evangelou, R. B. Schnabel, P. Welsh, P. Maffia, J. Erdmann, M. Tomaszewski, M. J. Caulfield, N. Sattar, M. V. Holmes, and T. J. Guzik. "White Blood Cells and Blood Pressure: A Mendelian Randomization Study." Circulation 141, no. 16 (2020): 1307-17.
  11. Wenzel, P., M. Knorr, S. Kossmann, J. Stratmann, M. Hausding, S. Schuhmacher, S. H. Karbach, M. Schwenk, N. Yogev, E. Schulz, M. Oelze, S. Grabbe, H. Jonuleit, C. Becker, A. Daiber, A. Waisman, and T. Munzel. "Lysozyme M-Positive Monocytes Mediate Angiotensin Ii-Induced Arterial Hypertension and Vascular Dysfunction." Circulation 124, no. 12 (2011): 1370-81.
  12. Kramer, A., J. Green, J. Pollard, Jr., and S. Tugendreich. "Causal Analysis Approaches in Ingenuity Pathway Analysis." Bioinformatics 30, no. 4 (2014): 523-30.

Reviewer 5 Report

Authors present a very interesting manuscript concerning the role of microRNA/WAVE2 in the pathogenesis of hypertension, its complications, as well as race related differences of molecular targets involved in hypertension. miRNA 1253 was previously analyzed as a possible target in glaucoma diagnosis (Allyson G Hindle, Robrecht Thoonen, Jessica V Jasien, Robert M H Grange, Krishna Amin, Jasen Wise, Mineo Ozaki, Robert Ritch, Rajeev Malhotra, Emmanuel S Buys. Identification of Candidate miRNA Biomarkers for Glaucoma. Invest Ophthalmol Vis Sci 2019 Jan 2;60(1):134-146. doi: 10.1167/iovs.18-24878).

In this paper Authors show results of microarray datamining to identify mRNA targets of miRNa 1253 in humans, later analyzed if miRNA 1253 can regulate UTR of WAVE2, and if miRNA 1253 expression is different in HAECs and HUVECs and changes WAVE2 expression. At the end a very interestinf finding that miRNA 1253 increases lamellipodia in HUVECs has been showed. Since Authors show novel data which may in part explain race differences in hypertension and its complications prevalence I think this manuscript is worth of publishing. Before that, please change the resolution of figures, Figure 2 is totally fuzzy, if possible please try improve figures 3, 4, 5B and increase the font of words on both axes. I guess the first part of the sentence starting in line 387 does not have a verb.

Author Response

Response to Review 5:

Dear Reviewer,

The authors thank you for providing a thorough and critical review our manuscript. We appreciate the opportunity to improve our work. Below, we have provided our responses and detailed the changes we have made to the manuscript to address these concerns. We thank the reviewer again for their time and hope this provides clarification.

Regards,

Doug Dluzen

  1. Authors present a very interesting manuscript concerning the role of microRNA/WAVE2 in the pathogenesis of hypertension, its complications, as well as race related differences of molecular targets involved in hypertension. miRNA 1253 was previously analyzed as a possible target in glaucoma diagnosis (Allyson G Hindle, Robrecht Thoonen, Jessica V Jasien, Robert M H Grange, Krishna Amin, Jasen Wise, Mineo Ozaki, Robert Ritch, Rajeev Malhotra, Emmanuel S Buys. Identification of Candidate miRNA Biomarkers for Glaucoma. Invest Ophthalmol Vis Sci 2019 Jan 2;60(1):134-146. doi: 10.1167/iovs.18-24878).

We thank the reviewer for identifying this useful paper and we have included it in our discussion of miR-1253’s roles in disease and health (lines 435).

  1. In this paper Authors show results of microarray datamining to identify mRNA targets of miRNa 1253 in humans, later analyzed if miRNA 1253 can regulate UTR of WAVE2, and if miRNA 1253 expression is different in HAECs and HUVECs and changes WAVE2 expression. At the end a very interestinf finding that miRNA 1253 increases lamellipodia in HUVECs has been showed. Since Authors show novel data which may in part explain race differences in hypertension and its complications prevalence I think this manuscript is worth of publishing. Before that, please change the resolution of figures, Figure 2 is totally fuzzy, if possible please try improve figures 3, 4, 5B and increase the font of words on both axes. I guess the first part of the sentence starting in line 387 does not have a verb.

We thank the reviewer for these comments. We have increased the fonts of the figures and made changes to Figures 1-5 per other reviewer comments as well. We have also increased the font size to make them more legible and fixed the typographical error that was missing. The font of the words on all of the graph axes has been increased for visibility.

We have adjusted Figure 2 so that it is larger. As the comparison between AAHT vs WHT is the most important, we have removed Part 2A in this figure and moved it Supplementary Figure 1A. This leaves Figure 2B as the sole diagram of the new Figure 2. We have also updated the Figure legend and corresponding results sections to reflect his modification.

Round 2

Reviewer 1 Report

The authors have made an effort to improve the manuscript. However, the main problem remains. The manuscripts have two differentiated parts.

First part is quite confusing, with a lot of gene lists in supplementary File 1, some of them obtained in a previous work. Even when there are a lot of lists, genes from the IPA analysis of transfected cells should also be there to facilitate the understanding of the different comparisons.

Based on the down-expression of miR-1253 in PBMCs from AAHT compared to those from WHT, and as it is resumed in Figure 1, authors started searching in silico targets for this miRNA (3088). Then, they compared this list with the compiled IPA gene set of hypertension-related genes and found 200 overlapping genes. These 200 genes were again compared with the genes with significant changes in expression levels in AAHT women (in this Figure 1C, “Targets differentially…” should read “Genes differentially…” because targets are in the other circle) and found 117 overlapping genes. These are in silico miR-1253 target genes, up or down regulated in PBMCs from AAHT women compared with WHT women, and hypertension-related. Authors say in lane 192-193 they did not filter these 117 predicted targets by expression directionality in hypertensive women, but, if you are looking for direct targets of a miRNA you should search for downregulated genes.

In any case, authors decided to over-express miR-1253 in HAECs to search for genes regulated for miR-1253 and they found 747, now, down regulated genes. Comparing these genes with the previous 117, they found 23 overlapping genes: in silico miR-1253 target genes, up or down regulated in PBMCs from AAHT women compared with WHT women, hypertension-related and repressed in HAECs after miR-1253 over-expression.

From here, it is difficult to understand the idea. Figure 1D shows 219 genes related to Cardiovascular Disease obtained from the transfection experiments, but in Supplementary File 1 there are 4114 genes in that category. Where are these genes came from?

After transfection analyses, authors go back to examine the list of Actin Cytoskeleton Signaling Pathway from the hypertension cohort. They found 27 from 75 genes significantly different comparing AAHT with WHT (Figure 2, please change in lane 238; and Supp. Table 1). What does it mean half red/half green in Figure 2 and Supp. Figure 1C? What do these genes mean? What was this comparison made for? Because the 23 previous genes were compared again with the 75 genes in the Actin Cytoskeleton Pathway (lane 277).

The conclusion of this first part is that only WASF2 gene is the candidate authors are looking for and they justify its role in hypertension in the discussion.

But the second part has nothing to do with this part. WASF2 is for sure a target of miR-1253, as authors demonstrate with clear transfection experiments. And, of course, WASF2 should have other regulators. In fact, miR-1253 cannot be implicated in the role of WASF2 in a hypertension context because both are down-regulated in that situation. It can be that another regulator is down-regulating WASF2 in hypertension, for example an upregulated miRNA, but authors do not look for it, or the down-regulation of miR-1253 is affecting another gene/protein and the consequence is down-regulating WASF2. The effects seen in the experiments of miR-1253 over-expression described here are effectively due to miR-1253 but are not reflecting the situation in a hypertension context, where miR-1253 is down-regulated. Thus, the conclusion that authors “identify that miR-1253 can regulate the WASF2 3’ UTR, repress WASF2 protein expression in endothelial cells, and influence actin cytoskeletal dynamics with respect to increased lamellipodia formation in endothelial cells” is accurate but has no relation with the hypertension context where the study started. Authors say that it would have been stronger data to see a reciprocal effect but the thing is that it would be the logical data.

Minor comments

1. Regarding Table 1, some p values in scientific notation and some not is very confusing. FLNA, PI3K and VCL should be bolded in AAHT vs WHT column. Also in this column, GSN should not be bolded. Please, confirm that are 26 instead of 27 significantly different genes.

2. In the Supplementary File 1 revised, actually, the list of 75 genes is not in the tab entitled “HAEC Target and Pathway Analysis” but in and independent tab entitled “Actin Cytoskeleton Pathway List” and also in the tab entitled “Cardiovascular Disease Pathway”.

3. If the synthetic miR-1253 you have used in the luciferase experiments is the same as in the other transfections, it is not the mimic you have transfected (lane 124) but the precursor. A mimic is a different product than a precursor and you should not change its name.

4. In Figure 5, it is unclear the presence of more lamellipodia in miR-1253 transfected cells. Maybe the picture is not representative because any filopodia is visible in control cells.

5. Text between lines 497 and 505 is crossed out and written again.

Author Response

Response to Reviewer 1:

Dear Reviewer,

The authors thank you for providing your input during the review process to improve our manuscript and for being patient with us. Below, we have provided our responses and detailed the changes we have made to the manuscript to address your concerns. We thank the reviewer again for their time.

Regards,

Doug Dluzen

The authors have made an effort to improve the manuscript. However, the main problem remains. The manuscripts have two differentiated parts.

1. First part is quite confusing, with a lot of gene lists in supplementary File 1, some of them obtained in a previous work. Even when there are a lot of lists, genes from the IPA analysis of transfected cells should also be there to facilitate the understanding of the different comparisons.

We thank the reviewer for their input. To help clarify the distinction from our prior work, we have updated the first paragraph of the Results section to acknowledge that while we had done very preliminary target prediction analysis in our 2016 study, we have since updated this analysis for our work presented here (lines 162-166). We have also updated the names of the tabs in the supplementary file so that they more clearly correspond with the Venn diagram comparisons in Figure 1A-C and the actin cytoskeleton pathway in Supplemental Table 1, as well as the 23 mRNAs listed in Table 1 in our comparison with the HAEC target analysis (tab called ‘Table 1 & HAEC Target Analysis’).

2. Based on the down-expression of miR-1253 in PBMCs from AAHT compared to those from WHT, and as it is resumed in Figure 1, authors started searching in silico targets for this miRNA (3088). Then, they compared this list with the compiled IPA gene set of hypertension-related genes and found 200 overlapping genes. These 200 genes were again compared with the genes with significant changes in expression levels in AAHT women (in this Figure 1C, “Targets differentially…” should read “Genes differentially…” because targets are in the other circle) and found 117 overlapping genes. These are in silico miR-1253 target genes, up or down regulated in PBMCs from AAHT women compared with WHT women, and hypertension-related. Authors say in lane 192-193 they did not filter these 117 predicted targets by expression directionality in hypertensive women, but, if you are looking for direct targets of a miRNA, you should search for downregulated genes.

We have corrected the wording in Figure 1C to clarify with the wording “differentially-expressed transcripts…” and we thank you for identifying this error. We understand your critique that looking for downregulated targets in AAHT and WHT women will help identify direct targets. Our analysis includes this. However, since many mRNAs are likely regulated by several mechanisms at once, including at times multiple miRNAs, we expanded our search filter to include transcripts up-regulated in order to capture potential interactions that are not likely just to be the sole interaction (but we agree, miR-1253 regulation of these transcripts may not be a primary mechanism). This approach is meant to support our in silico analysis and not be conclusive proof all of these targets are regulated by miR-1253. We have updated our results section to clarify on this point and further explain this reasoning (lines 184-185).

3. In any case, authors decided to over-express miR-1253 in HAECs to search for genes regulated for miR-1253 and they found 747, now, down regulated genes. Comparing these genes with the previous 117, they found 23 overlapping genes: in silico miR-1253 target genes, up or down regulated in PBMCs from AAHT women compared with WHT women, hypertension-related and repressed in HAECs after miR-1253 over-expression. From here, it is difficult to understand the idea. Figure 1D shows 219 genes related to Cardiovascular Disease obtained from the transfection experiments, but in Supplementary File 1 there are 4114 genes in that category. Where are these genes came from?

Thank you for identifying this error. We had mistakenly copied in the full list of genes in this category from IPA instead of the 219 molecules identified in the IPA analysis of our data and referenced in Figure 1D. When we started to correct this, we realized this was a good opportunity to combine the 219 gene list with that of the 521 molecules in the Cellular Growth and Proliferation and the 260 molecules in the Cellular Organization and Assembly which had been in bold, previously, in Figure 1D but we had not previously discussed in any detail. We did this to be more comprehensive and identify the total number of genes in the actin cytoskeleton pathway that are found in the top pathways in IPA. After removal of duplicate genes within these three lists, we were left with 750 genes, which are listed in the tab ‘Fig1D_Molecules in IPA’. Comparison of this new list with the genes in the actin cytoskeleton pathway identified the 16 genes we had previously mentioned in the Results section of the manuscript, and an addition 7 in this pathway, bringing the total to a final 23 genes (and not to be confused with the 23 genes identified prior in the analysis and listed in Table 1 which are not necessarily involved with the actin cytoskeleton). All of these comparisons and lists are clearly delineated in the Suppl File 1.

We have also updated our Results section (lines 212-218) to clarify on this change and it now reads as:

“We next examined DGE in the actin cytoskeleton signaling pathway in our hypertension cohort by reanalyzing our previous microarray dataset GSE75672 from [9] (gene list in Supp. File 1). We chose this pathway for several reasons. First, given the role of actin cytoskeletal remodeling and signaling in hypertension and endothelial function [29-31]. Second, given the importance of this pathway in the Cardiovascular Diseases, Cellular Growth and Proliferation, and Cellular Assembly and Organization categories identified in our IPA analysis (Figure 1D, bold), in which 23 genes in the actin cytoskeleton pathway are found in the combined molecules list in IPA after removal of overlapping molecules (see Supp. File 1 for full list of molecules and overlapping genes). Third, our prior analysis that RHOA, a member of the actin cytoskeleton pathway and associated with hypertension etiology in endothelial and vascular smooth muscle cells, is differentially expressed in PBMCs between AA and white hypertensive women. And fourth, the fact that this particular pathway in PBMCs is relatively unexplored with respect to hypertension.”

4. After transfection analyses, authors go back to examine the list of Actin Cytoskeleton Signaling Pathway from the hypertension cohort. They found 27 from 75 genes significantly different comparing AAHT with WHT (Figure 2, please change in lane 238; and Supp. Table 1). What does it mean half red/half green in Figure 2 and Supp. Figure 1C? What do these genes mean? What was this comparison made for? Because the 23 previous genes were compared again with the 75 genes in the Actin Cytoskeleton Pathway (lane 277).

Using IPA, you can overlay gene expression datasets into canonical pathways to visually represent what is happening in the entire pathway. If a gene is represented as half green/half red, this is an IPA-designated indication that these genes bind with each other or have some important functional interaction together in the pathway. For example, in Figure 2, the VAV/TIAM icon is both red and green in the picture. In Suppl. Table 1, VAV is upregulated while TIAM is down, thus it shows both red and green. In the case of PI3K, in which only one gene name is listed, IPA’s default settings will collapse values from transcript variants of that gene to represent that gene overall trend. This is what is reported in Suppl Table 1. While for the very few nodes this occurs, it’s meant to simply the pathway picture, as showing all transcript variants for each gene, such as PI3K, would be impractical. However, should someone be interested in a specific transcript, all values for all individual transcript variants can be found in our Gene Expression Omnibus dataset that is open online. We have updated the Figure 2 legend to clarify this and indicate why the icons may be bi-colored (lines 245-248).

To address your question of ‘What was this comparison made for?’: We compared gene expression in the actin cytoskeleton in PBMCs in AAHT vs WHT to see how this entire pathway is behaving in PBMCs. We find that 27 out of the 75 actin genes are differentially-expressed in these two groups. As it states in our manuscript, we were next interested in seeing how many of the miR-1253 targets are also differentially-expressed in AAHT vs WHT. We used our list of 747 mRNAs (down-regulated when miR-1253 is expressed in HAECs) as a starting point. We find that 23 of these genes are also differentially-expressed between AAHT vs WHT. If these genes weren’t differentially-expressed between AAHT and WHTs, we weren’t interested further in pursuing them for this study. We next compared these 23 genes with the 75 in the actin cytoskeleton pathway and the only gene that overlaps is WASF2. We did the comparison in this way to show that while there are several (23) miR-1253 targets down-regulated in HAECs and also differentially-expressed in PBMCs, only one is found in the actin cytoskeleton. That is why we focus on this gene in particular. This is the first time in the manuscript that actin cytoskeleton genes are directly compared with hypertension-related genes that are also down-regulated by miR-1253 over-expression. We made some minor changes in the results to help clarify this even more (lines 257-258).

We acknowledge that this approach leaves the other 22 genes unexplored at this time. However, given our interest in actin cytoskeleton signaling, this was our work flow.

5. The conclusion of this first part is that only WASF2 gene is the candidate authors are looking for and they justify its role in hypertension in the discussion. But the second part has nothing to do with this part. WASF2 is for sure a target of miR-1253, as authors demonstrate with clear transfection experiments. And, of course, WASF2 should have other regulators. In fact, miR-1253 cannot be implicated in the role of WASF2 in a hypertension context because both are down-regulated in that situation. It can be that another regulator is down-regulating WASF2 in hypertension, for example an upregulated miRNA, but authors do not look for it, or the down-regulation of miR-1253 is affecting another gene/protein and the consequence is down-regulating WASF2. The effects seen in the experiments of miR-1253 over-expression described here are effectively due to miR-1253 but are not reflecting the situation in a hypertension context, where miR-1253 is down-regulated. Thus, the conclusion that authors “identify that miR-1253 can regulate the WASF2 3’ UTR, repress WASF2 protein expression in endothelial cells, and influence actin cytoskeletal dynamics with respect to increased lamellipodia formation in endothelial cells” is accurate but has no relation with the hypertension context where the study started. Authors say that it would have been stronger data to see a reciprocal effect but the thing is that it would be the logical data.

We have indicated in the Discussion that we believe there could be a connection with hypertension, as these genes are differentially-expressed in hypertensives, but that more work is needed clarify on a cause or consequence. We have re-organized a part of the discussion and added more discussion in a new paragraph on the study limitations to drive home the point these genes should be considered in future studies related to the etiology of hypertension. This can be found in lines 461-467 and reads:

“Our approach was useful in being able to examine DGE patterns in understudied populations and identify novel regulators of some of those genes. But our study is limited because it is not known whether differential-expression of miR-1253 or WASF2 in AA women with hypertension is a contributing cause or a consequence of elevated high blood pressure. It is likely not a major cause of hypertension development, but given the pathways these genes are involved with, they should be explored further when it comes to understanding how to better manage or treat high blood pressure. We were also not able to ascertain if miR-1253 is the major regulator of WASF2 transcript levels in PBMCs or acting as a contributing role player. As the connections between circulating cells and the development and pathology of hypertension become clearer, these genes should be considered. The miR-1253-WASF2 interaction may also be relevant, or perhaps even more relevant, in tissues already known to be more directly related to the constriction and dilation of the vasculature, including the endothelial and smooth muscle layers. Our finding highlights the need to validate if miR-1253 regulates WASF2 expression in primary endothelial or VSMCs, particularly in projects focused on examining mechanisms of hypertension-related disparity and in biospecimens from these populations. Given there is no data in the literature examining whether changes in miR-1253 influence endothelial dysfunction in response to increased blood pressure, our study lays groundwork for these intriguing projects.”

Minor comments

  1. Regarding Table 1, some p values in scientific notation and some not is very confusing. FLNA, PI3K and VCL should be bolded in AAHT vs WHT column. Also in this column, GSN should not be bolded. Please, confirm that are 26 instead of 27 significantly different genes.

We believe the reviewer was referring to Supplementary Table 1. We have put all P-values into scientific notation for all genes and all comparisons. We have bolded in FLNA, PI3K, and VCL in the required column, as they are significant. We have also removed the Bold on GSN. By our count, the bold of GSN was an additional typo, and even after removal, there are still 27 significantly-different genes in the AAHT vs WHT comparison.

  1. In the Supplementary File 1 revised, actually, the list of 75 genes is not in the tab entitled “HAEC Target and Pathway Analysis” but in and independent tab entitled “Actin Cytoskeleton Pathway List” and also in the tab entitled “Cardiovascular Disease Pathway”.

We had thought to include the Actin Cytoskeleton Pathway List in the Cardiovascular Disease Pathway tab as this would help provide where we conducted our overlap. However, we realize this can cause confusion as the reviewer has pointed out. We have deleted this from the ‘Cardiovascular Disease Pathway’ tab and left it as a separate tab now labeled ‘SupplTable1_ActinCytoskeleton’ so that it isn’t confused with anything else and also refers to the genes listed in Supplementary Table 1.

  1. If the synthetic miR-1253 you have used in the luciferase experiments is the same as in the other transfections, it is not the mimic you have transfected (lane 124) but the precursor. A mimic is a different product than a precursor and you should not change its name.

We have changed the name of the miR-1253 mimic in the luciferase methods section to be consistent, as we had used the precursor mimic as defined early in the methods (line 121).

  1. In Figure 5, it is unclear the presence of more lamellipodia in miR-1253 transfected cells. Maybe the picture is not representative because any filopodia is visible in control cells.

The pictures in Figure 5A are meant to be representative of one the most typical pictures for each of the scrambled control (left) or miR-1253 mimic (right). We chose these photographs as they represent more lamellipodia (white arrows) compared with fewer in scrambled control and also depicted a filopodia structure as a comparison of what lamellipodia look like.  As these are meant as a visual for our scoring, the quantification of all our cells in panels B and C are the result of scoring all of our cells and images. If we zoomed out on an image to visually capture the increase in lamellipodia, we’d lose out on the definition to see each structure. We feel this is the best way to present our data. We have made minor changes to the Figure 5 legend, lines 346-347, to indicate this as an example picture more clearly.

  1. Text between lines 497 and 505 is crossed out and written again.

Thank you for pointing this out to us. We believe this is an issue just with the track changes. In the document with accepted changes, this disappears.

Reviewer 3 Report

The authors have addressed most of the points raised by the reviewer and the manuscript has improved considerably. The analysis shown in Figure 1 in the revised version is now explained more clearly and, importantly, the 117 genes identified by this analysis are later on compared with the set of downregulated genes found upon miR-1253 mimic overexpression. Also, it is now more clearly understood why the authors focused on the actin cytoskeleton signaling pathway, and the list of genes from this signaling pathway deregulated upon miR-1253 mimic overexpression is provided as a supplementary material in the updated version. It is unfortunate that the authors could not include a rescue experiment overexpressing WAVE2 together with miR-1253 mimic, but this reviewer is well aware of the current extraordinary circumstances due to the pandemic. The paper is now acceptable for publication.

Please, check the following minor points:

  1. In the new Figure 1C, please specify in the Venn diagram “miR-1253 target genes related wit hypertension” as only the 200 genes obtained from 1B are compared in 1C.
  2. Please, carefully proofread, some typos can be found, for example in lines 186 and 221.

Author Response

Response to Review 3:

Dear Reviewer,

The authors thank you for providing your input during the review process to improve our manuscript. Below, we have provided our responses and detailed the changes we have made to the manuscript to address these concerns. We thank the reviewer again for their time.

Regards,

Doug Dluzen

The authors have addressed most of the points raised by the reviewer and the manuscript has improved considerably. The analysis shown in Figure 1 in the revised version is now explained more clearly and, importantly, the 117 genes identified by this analysis are later on compared with the set of downregulated genes found upon miR-1253 mimic overexpression. Also, it is now more clearly understood why the authors focused on the actin cytoskeleton signaling pathway, and the list of genes from this signaling pathway deregulated upon miR-1253 mimic overexpression is provided as a supplementary material in the updated version. It is unfortunate that the authors could not include a rescue experiment overexpressing WAVE2 together with miR-1253 mimic, but this reviewer is well aware of the current extraordinary circumstances due to the pandemic. The paper is now acceptable for publication.

We thank the reviewer for these comments and their additional input. We have responded to the two minor points below.

Please, check the following minor points:

  1. In the new Figure 1C, please specify in the Venn diagram “miR-1253 target genes related wit hypertension” as only the 200 genes obtained from 1B are compared in 1C.

We have updated Figure 1 C above the 200 genes in the Venn diagram to ‘miR-1253 target genes related with hypertension’ to clarify.

2. Please, carefully proofread, some typos can be found, for example in lines 186 and 221.

We have read through the revised manuscript to address any typos. We have also identified the two brought to our attention here and corrected them, as well as made a few other additional minor typographical fixes. Thank you for identifying these.

Reviewer 4 Report

see separated mail.

Author Response

Response to Review 4:

Dear Reviewer,

The authors thank you for providing your input during the review process to improve our manuscript. Below, we have provided our responses and detailed the changes we have made to the manuscript to address these concerns. We thank the reviewer again for their time and this help critique!

Regards,

Doug Dluzen

Review for the authors:

The authors have chosen for a more modest (limited) presentation, as already from the title can be deduced. I think this is wise and reflects more properly the scope of their research.

Also, they make their link with the previous paper (Dluzen et al. 2016) more direct and clear. Again an improvement. All in all, I apreciate the (textual) changes the authors have made.

I understood and understand that starting from PMBCs is easier, in particular if one has to collect so many samples from different groups of patients. However, I still think that gene expression levels in PMCBs have only a marginal relation with bloodpressure, and therefore I stick to my point of view that the genes selected from these cells are not a good representation of the genes involved in hypertention. Have stated that, I am sensitive to the argument in your responds, that obtaining smooth muscle cells of blood vessels (with the question, which blood vessels) and/or endothelian cells from so many patients is an almost impossible task. This calls for a consortium of many hospitals/labs collaborating to set up such a collection of tissues/cells.

Although the discussion has improved considerably, I would like to make a suggestion. Spread over the discussion there are leads/suggestions for further investigations and also remarks on the limitations of this study. In particular for the limitations of this study I would prefer a (small) section, pointing out what the range of the described findings is and how they can be further explored. This section would also offer space to discuss whether the observed observations are causal or a consequence of hypertension.

We thank the reviewer for their comments and helpful input. We have reorganized some of the discussion and feature a new paragraph with additional discussion on the role and range of these findings, and focusing on important take-home points as well as limitations of our study. This is in lines 461-477 and featured below:

“Our approach was useful in being able to examine DGE patterns in understudied populations and identify novel regulators of some of those genes. But our study is limited because it is not known whether differential-expression of miR-1253 or WASF2 in AA women with hypertension is a contributing cause or a consequence of elevated high blood pressure. It is likely not a major cause of hypertension development, but given the pathways these genes are involved with, they should be explored further when it comes to understanding how to better manage or treat high blood pressure. We were also not able to ascertain if miR-1253 is the major regulator of WASF2 transcript levels in PBMCs or acting as a contributing role player. As the connections between circulating cells and the development and pathology of hypertension become clearer, these genes should be considered. The miR-1253-WASF2 interaction may also be relevant, or perhaps even more relevant, in tissues already known to be more directly related to the constriction and dilation of the vasculature, including the endothelial and smooth muscle layers. Our finding highlights the need to validate if miR-1253 regulates WASF2 expression in primary endothelial or VSMCs, particularly in projects focused on examining mechanisms of hypertension-related disparity and in biospecimens from these populations. Given there is no data in the literature examining whether changes in miR-1253 influence endothelial dysfunction in response to increased blood pressure, our study lays groundwork for these intriguing projects.